# Modelling the evolution of Arctic multiyear sea ice over 2000-2018

Heather Regan[1], Pierre Rampal[2,1], Einar Ólason[1], Guillaume Boutin[1], and Anton Korosov[1]

[1]Nansen Environmental and Remote Sensing Center, and the Bjerknes Center for Climate Research, Bergen, Norway
[2]CNRS, Institut de Géophysique de l'Environnement, Grenoble, France

**Correspondence:** Heather Regan (heather.regan@nersc.no)

**Abstract.** Multiyear sea ice (MYI) cover in the Arctic has been monitored for decades using increasingly sophisticated remote sensing techniques, and these have documented a significant decline in MYI over time. However, such techniques are unable to differentiate between the processes affecting the evolution of the MYI. Further, estimating the thickness, and thus the volume of MYI remains challenging. In this study we employ a sea ice-ocean model to investigate the changes to MYI over the period 2000-2018. We exploit the Lagrangian framework of the sea ice model to introduce a new method of tracking MYI area and volume, which is based on identifying MYI during freeze onset each autumn. The model is found to successfully reproduce the spatial distribution and evolution of observed MYI extent. We discuss the balance of the processes (melt, ridging, export, and replenishment) linked to the general decline in MYI cover. The model suggests that rather than one process dominating the losses, there is an episodic imbalance between the different sources and sinks of MYI. We identify those key to the significant observed declines of 2007 and 2012; while melt and replenishment are important in 2012, sea ice dynamics play a significant role in 2007. Notably, the model suggests that in years such as 2007, convergence of the ice, through ridging, can result in large reductions of MYI area without a corresponding loss of MYI volume. This highlights the benefit of using models alongside satellite observations to aid interpretation of the observed MYI evolution in the Arctic.

## 1 Introduction

Arctic sea ice has undergone a significant decline (e.g., Comiso, 2012) and thinning (Kwok and Rothrock, 2009) in recent decades, with subsequent impacts on climate, biodiversity and human activities in the region (IPCC Special Report on the Ocean and Cryosphere in a Changing Climate, Meredith et al., 2019). Remote sensing studies suggest that the multiyear ice (MYI) - ice that has survived at least one summer - has been experiencing a similar, if not more rapid decline than the total ice cover and younger ice (Comiso et al., 2008; Comiso, 2012), with an almost complete loss of ice older than 5 years (Maslanik et al., 2007) compared to such ice types making up 50% of the MYI cover in the central Arctic in the 1980s (Maslanik et al., 2011). During the period 2003-2008, overall ice volume and thickness losses were found to be predominantly due to changes to MYI (Kwok et al., 2009). As such, the Arctic ice cover is now dominated by seasonal ice (Kwok, 2018).

Studying the evolution of MYI in the Arctic is important for understanding how the overall ice pack is changing, as well as for quantifying the associated feedbacks in the system. As ice ages, freezing and ridging lead to a direct relationship between large-scale ice age and thickness (Tschudi et al., 2016; Hunke and Bitz, 2009; Maslanik et al., 2007), with MYI area anomalies being shown to be closely linked to anomalies in Arctic ice volume (Kwok, 2018). This relationship is important, both for

inferring interannual variability in ice volume from ice age data, and for understanding the processes driving changes to the evolving ice pack. Freezing and ridging lead to changes to the surface properties of the ice cover, resulting in a higher albedo than that of FYI (Perovich et al., 2003). At the same time, the correlation between ice age and salinity due to desalination over time affects the thermal properties of the sea ice as well as the ice–ocean salt and freshwater exchanges, thus influencing the sea ice mass balance and the characteristics of the polar oceans (Vancoppenolle et al., 2009a, b). A reduction of the MYI cover indicates a thinning of Arctic sea ice, which in turn increases the ice-free portion of the Arctic in the summer (Holland et al., 2006), leading to positive feedbacks and changes to the surface ocean (Haine and Martin, 2017). The MYI cover in the Arctic can thus provide an indication of both the thickness of the Arctic ice cover, and its ability to withstand anomalous or changing forcing.

Much of the knowledge of the pan-Arctic MYI distribution has been gained from studies of satellite observations, often in conjunction with in-situ data (e.g., Fowler et al., 2004; Rigor and Wallace, 2004; Maslanik et al., 2011). These generally exploit some of the key differences in near-surface properties of first-year ice (FYI) and MYI to determine ice types in a particular pixel (e.g., Kwok, 2004; Aaboe et al., 2019), or use ice concentration in conjunction with ice motion to track pixels of ice and age them at the end of summer (e.g., Rigor and Wallace, 2004; Korosov et al., 2018). To date, such studies have provided a large amount of valuable information on the MYI extent or area in the Arctic, such as the dramatic losses of over 50% of MYI from 1999 to 2017 (Kwok, 2018) and depletion of older ice types (Maslanik et al., 2011).

Satellite observations of ice types and age still have several inherent or hard-to-overcome limitations. Ice type classification is more difficult in summer, due to surface melt (e.g., Kwok, 2004; Aaboe et al., 2019), and therefore some products limit availability to just the winter months. At the same time, methods that use ice motion must make assumptions about when and how to age pixels, and must rely on ice drift products which are improving but have historically been significantly less reliable in summer (see Maslanik et al., 1998; Lavergne et al., 2010; Sumata et al., 2014). Besides these shortcomings, information about the role of different processes involved in the evolution of MYI is generally difficult to obtain from satellites. This is particularly true for melt and replenishment (conversion of FYI to MYI), as well as ridging. Melt and replenishment occur at a time of year when satellite information is least reliable (e.g., Kwok and Cunningham, 2010), while distinguishing the area changes due to ridging relies on accurate ice drift (von Albedyll et al., 2021) and therefore assessing its contribution in summer is challenging. Finally, ice thickness information with good spatial and temporal coverage has only been available from satellite altimetry since the early 2000s (e.g., Zygmuntowska et al., 2014), with year-round information only being available very recently (Landy et al., 2022). Information about the evolution of MYI volume is, therefore, difficult to ascertain from observations.

In this study, we assess the budget of MYI and its interannual evolution. While previous modelling approaches have tended to focus on ice age as a tracer (e.g., Hunke and Bitz, 2009; Hunke, 2014; Vancoppenolle et al., 2009b), here we implement tracers of MYI concentration and volume in a coupled setup of the neXtSIM sea ice model (Rampal et al., 2016) in its latest version (Ólason et al., 2022) with the ocean component (OPA) of the NEMO (Nucleus for European Modelling of the Ocean) model (Madec, 2008). We exploit the Lagrangian framework in neXtSIM to introduce a new method to track the evolution of MYI. In our implementation we track MYI concentration, having an end-of-summer MYI source term based on the local

autumn freeze onset as opposed to an arbitrary date, in a manner more directly comparable to the surface signature of MYI from satellites. Combining this with the tracking of MYI volume, we can then explore how changes at the surface manifest in thickness.

The paper is structured as follows. Section 2 outlines the model setup and introduces definitions and details of the MYI tracers implemented in the model. Section 3 provides a thorough evaluation of the modelled MYI extent against a MYI type dataset derived from satellite observations. Section 4 then provides an analysis of the MYI volume and a partitioning of the processes contributing to its interannual variability, and additionally focuses on two extreme years and how these contribute to the interannual decline. Section 5 discusses anomalous conditions in the context of the available literature, and how the

definition of MYI volume can affect its evolution. Section 6 summarises and concludes.

## 2   Methods

### 2.1   Model setup

We use an ice-ocean coupled simulation to study the MYI in the Arctic. The ocean component is that of the Nucleus for European Modelling of the Ocean (NEMO) model, version 3.6. We use the CREG configuration (Dupont et al., 2015), a

seamless regional extraction of the configuration developed by the Drakkar consortium and Mercator Ocean (Barnier et al., 2006), containing the full Arctic Ocean and extending down to 27°N in the Atlantic. We modify the original configuration to use the neXt generation Sea Ice Model (neXtSIM; Rampal et al., 2016) in place of the NEMO sea ice model, LIM3. The neXtSIM model differs from LIM3 and most of the traditional sea ice models in that it uses the brittle Bingham-Maxwell rheology (Ólason et al., 2022) to represent sea ice mechanics. It also differs in that it uses a pure Lagrangian advection framework,

meaning that the mesh on which the model runs is being distorted and adapted throughout the simulation according to the computed ice motion. The Lagrangian framework is particularly well suited for implementing the advection of multiple sea ice tracers.

    The setup is run at 0.25° horizontal resolution (Talandier and Lique, 2021), resulting in a grid spacing in the Arctic of around 12 km, with 75 vertical levels in the ocean. The model uses monthly climatological boundary conditions at the Bering Strait

and the southern boundary from a longer ORCA025 simulation performed by the Drakkar Group, and initial conditions are taken from the *World Ocean Atlas 2009*. The model is forced by the hourly ERA5 reanalysis (Hersbach et al., 2020), with an updated version of river and ice sheet runoff from Dai and Trenberth (2002) which includes the increasing contribution from Greenland (Gillard et al., 2016). The model variables are output every 6 hours and the simulation covers the period from 1995-2018. We present results from 2000 onwards to allow for spinup of the sea ice. More details of the coupled neXtSIM-OPA

model setup and an evaluation of the sea ice properties can be found in Boutin et al. (2023). In brief, sea ice extent, volume and drift are found to agree generally well with observations, especially for the drift. Sea ice volume export through Fram Strait - particularly its variability - is also consistent with observed estimates, but may be underestimated prior to 2008 (the observations show large uncertainties). Boutin et al. (2023) also compared their dynamic and thermodynamic components of

the winter mass balance against estimates by Ricker et al. (2021) for the period 2003–2018. The model shows a reasonable
match for the thermodynamics, and is able to capture the variability of the dynamic changes in sea ice volume.

## 2.2  Model definition of MYI

MYI is ice that has survived at least one summer, while FYI is ice that has formed since the last summer melt season (e.g.
Carsey, 1992). Classification of these sea ice types from satellite observations is generally undertaken by exploiting the differ-
ences in surface and near-surface properties of the sea ice cover that occur as ice ages. When ice forms, it is generally smooth
and relatively saline. When it undergoes significant melting, it loses brine, leaving behind pockets of air in the ice cover and
becoming less salty, which affects its emissivity (Vant et al., 1978). Changes to surface roughness, for example through ridging
and snowfall, and developments of inhomogeneities in the increasingly low-salinity ice also alter its backscatter (Kwok et al.,
1992, 1999). Thus, changes to the physical properties of the ice result in key differences between newly formed FYI and older
MYI that are seen as distinct signatures in both passive and active satellite sensors, allowing the two ice types to be identified
and analysed. By the end of summer, by definition the FYI that is left after undergoing these physical changes becomes MYI. In
order to reflect this transition from FYI to MYI in the model, we tag ice concentration and volume left at the end of summer as
MYI concentration and MYI volume, respectively. Thus, in the model, newly formed ice is classified as FYI until it undergoes
the melt season, after which it is 'upgraded' to MYI.

A question that arises is how to identify the 'end of summer'. Studies of ice age that track ice parcels from satellites and
age them at the end of summer often use the minimum ice extent, or simply the month of September to determine this date
(e.g., Rigor and Wallace, 2004). Tracking the summer minimum extent is difficult to do in real time in the model. However, as
Zwally and Gloersen (2008) state, choosing the summer minimum extent or area as the date at which to upgrade MYI ignores
regional variability in the end of the melt season and can therefore erroneously include some regions that have already begun
to refreeze, leading to an overestimation of the MYI. It also does not account for any interannual variability in the freeze onset.
To determine the end of summer from a physical perspective in the model at a regional scale, we use the 'thermodynamic ice
growth tendency' condition to identify freeze onset that was examined by Smith and Jahn (2019). This condition states that
the onset of freezing following the summer melt has occurred in a grid cell element if $n$ number of consecutive days of net
ice growth have occurred since the summer melt began. As in Smith and Jahn (2019), we use $n = 3$, noting that tests of $n = 5$
and $n = 10$ yielded qualitatively similar results. In the model, we begin the check on summer melt onset for each grid cell on
1[st] August onwards each year, which ensures we do not capture intermittent spring freezing but do not miss the start of the
autumn refreezing. Basing the upgrade of FYI to MYI on a physical condition means that it is sensitive to interannual and
spatial variability in the freeze conditions, which is therefore more in line with the physical processes that result in different
signatures detected from satellites.

Figure 1 demonstrates the spatial variability of the freeze onset day in the model. The 2000-2018 average freeze onset day
varies spatially (Figure 1a), with the central Arctic north of the Canadian Arctic Archipelago and Greenland starting to refreeze
before day 250 (the 7[th] September), and the bulk of the ice cover starting to refreeze by mid-October. There is also a notable
interannual variability in the freeze onset day at each location; north of Greenland, for example, the earliest date of the onset

out of all simulation years occurs in early August (white-blue region in Figure 1b), with much of the ice pack experiencing at least one year where refreezing begins before the end of August (before day 240, in the light blue region). By contrast, the spatial distribution of the latest onset of refreezing is much more varied (Figure 1c), with the ice pack in the central Arctic refreezing as late as day 270 (the 27th September) and the regions of thinner ice and shelf seas refreezing as late as November or December. In general, the refreezing of locations of thicker ice vary by up to 30 days during the simulation, while regions of thinner ice can vary by over two months. This highlights the importance of using a spatially and temporally varying upgrade to MYI to avoid capturing newly formed FYI.

## 2.3 Source and sinks of MYI

In the model we trace MYI volume and concentration as the fraction of total volume and concentration in each element of the mesh. The FYI volume and concentration are calculated as the difference between the total volume and concentration and the MYI terms. MYI and FYI volume and concentration are affected by freezing and melting, by replenishment (the upgrade of FYI to MYI), by ridging, and at the basin scale by export. These processes are treated as follows:

– Freezing contributes to increase of FYI volume and concentration only.

– Melting acts as a sink term for both FYI and MYI concentration and volume. In the model, melt is calculated for the total concentration and volume, and we assume that MYI and FYI melt at the same rate. By doing so, we implicitly assume that the FYI has grown to the same thickness as the MYI by the end of the growth season and that it remains the case during the melting season. This is likely an upper bound for the melt rate of MYI, for 3 main reasons. First, MYI being generally thicker than FYI, the areal melt rate of FYI should be larger (as neXtSIM assumes this melt rate decreases with sea ice thickness, Rampal et al., 2016). Second, MYI generally has a thicker snow cover than FYI, which should delay the melt. Third, in this study, we consider that ice growth due to basal freezing under MYI is not a source term of MYI, and is therefore a source of FYI (see our discussion section 5.3). When the melt season starts, the MYI as we define it should only melt after all this FYI at the bottom has disappeared, which we do not factor in with our assumption on the MYI melt.

– Replenishment of MYI occurs when the ice in a mesh element has undergone three consecutive days of mean growth, following the height of the melt season (set as August 1$^{st}$). Upon replenishment, the MYI concentration and volume tracers are set equal to the total ice concentration and volume at the beginning of the freeze onset (i.e., three days earlier).

– Convergence, through ridging, of ice acts as a sink term for area only, not affecting volume. In the model, we assume that MYI ridges only after all FYI is ridged. In practice, convergence causes the triangular elements of the mesh to become smaller. If the element is fully ice covered, then ridging occurs. When this happens, the model simply assumes that FYI area is reduced (if there is any) and the area of MYI is conserved, as long as this area remains smaller than the area of the triangle. If this is not the case, MYI ridges and its area is reduced. This choice is based on our expectation that MYI

is generally both thicker and stronger than FYI and so nearly all, if not all, ridging should take place within the FYI area, as long as this exists. This hypothesis is source of uncertainty as it may underestimate MYI ridging in areas where MYI and FYI have fairly similar thickness (like the MIZ or in the Transpolar Drift). It may also be less representative in recent years, as the observed MYI thinning is faster than the one of FYI (e.g. Kwok, 2018).

     – MYI export is computed as integrated area and volume of the MYI fraction in the elements leaving a given region, which
we compute for the full Arctic and selected sub-regions (section 4).

## 2.4 Observations of ice type

We use version 1 of the sea ice type Climate Data Record from the Copernicus Climate Change Service (C3S) Climate Data Store (CDS) (2020), hereafter CDR, for comparison with the model, both to gain initial verification of our choice of how to define and evolve MYI, and then for an extensive analysis of simulated MYI extent (section 3). The daily dataset spans October
to April from 1979-present, allowing for a comparison from autumn to spring over the time period that we run the model for. The dataset primarily uses brightness temperatures from passive microwave radiometers to classify ice types on a 25 km grid covering the Northern Hemisphere. It uses a Bayesian approach to obtain the probability of the signature being a given surface type (MYI, FYI or open water); any pixel that has more than 30% ice concentration and 75% probability of being MYI is classified as MYI. Any grid cell that has less than 75% probability of being either MYI or FYI is classified as ambiguous. The
total extent of MYI in the Arctic from the CDR data is shown as the black line in Figure 2a. Further details of the satellite data and classification method are provided in Aaboe et al. (2019). The Arctic domain we use in our evaluation (Figure 2c) is the region bounded by the Pacific, Canadian Arctic Archipelago, Fram Strait and Barents Sea gateways, similar to the region used in Kwok (2018).

     The modelled MYI concentration is converted to a binary type classification for comparison with the CDR dataset using a
threshold $T$: elements with MYI concentration exceeding $T$ are considered as MYI. The extent of modelled MYI (area of the elements detected as MYI) is sensitive to the value of $T$. Figure 2a shows the evolution of MYI extent within the Arctic (Figure 2c) computed with values of $T$ ranging from 0.30 (blue) to 0.70 (red). The optimal value of $T$=0.40 was found by minimization of the root mean square difference (RMSD) between the MYI extent from CDR and from the model (Figure 2b). RMSD was computed from November to March due to large uncertainties in other periods; this affected the magnitude of the RMSD but
not the value of $T$. To have an idea of the uncertainty of the CDR dataset, we use the "uncertainty" variable provided in the dataset and assign the lower bound of CDR MYI as only those grid cells with an uncertainty $< 2\%$. We assign the upper bound of CDR MYI as all MYI cells plus ambiguous cells.

## 2.5 Observations of ice age

For additional evaluation of our model, we use the dataset of Arctic ice age from the National Snow and Ice Data Center
(NSIDC) (Tschudi et al., 2019), hereafter NSIDC_age. The dataset is constructed by creating a 12.5 × 12.5 km grid of ice parcels for each age category, which are then advected and tracked as Lagrangian parcels according to weekly sea ice motion

vectors estimated from satellite observations (Maslanik et al., 2011; Tschudi et al., 2020) and then interpolated back onto the original grid. If two parcels of different ages are advected into the same grid cell, the oldest age is taken for that grid cell, under the assumption that younger ice is easier to deform than older ice. At the summer minimum ice extent, if parcels remain, they are put into the next age category. More information can be found in Tschudi et al. (2016, 2019, 2020). The approach taken to produce this dataset is rather different to that of the CDR ice type: it is based on the age of ice, rather than type, which makes a direct comparison of the two datasets difficult. Further, by choosing the oldest ice in the case formulated above, it may overestimate the age of the ice by prioritising small concentrations of old ice (Korosov et al., 2018). Despite differences between these methods, the dataset can still provide some information on how well the model is advecting the ice, particularly at lower MYI concentrations which the CDR data does not provide. Additionally, it provides weekly data during the summer season when satellites that differentiate between ice types based on surface properties fail. To compare with ice types from our model, we assign ice types to the NSIDC_age product as FYI (ice age less than one year) and MYI (ice greater than one year). A similar approach was used for a comparison of this dataset with MYI in CESM previously (Jahn et al., 2012). We show the total extent of these age-based types on Figure 3a.

## 3   Evaluation of modelled MYI

Our main evaluation of the modelled MYI concentration is against the CDR data. We use the NSIDC_age data when no CDR data is available and to provide an indication how the results of different observational products may vary. We first consider the evolution of total MYI extent, comparing the model results against the two data sets in Figure 3a. The CDR data exhibits a substantial amount of noise in the daily values, but this is generally within the uncertainty estimate of the product. We also note that the MYI extent in the CDR data increases sharply during the autumn in some years, and may continue increasing throughout winter in others. While such an increase in MYI extent can occur mid-winter due to diverging ice drift, neither the NSIDC_age data nor the model show such an extensive increase. Note that in this study we use version 1 of the dataset, which is now deprecated and fully replaced by an upgraded version. The recently released version 3 of the dataset has major improvements relative to version 1, including both gap filling (bringing it to a level 4 product) and correction schemes based on temperature and ice drift information to correct for mis-classifications and to improve the early autumn classification (S. Aaboe, *personal comms*).

In general, the correspondence between the three datasets is reasonable. There are no obvious biases between them, beyond the fact that MYI extent derived from the NSIDC_age data is generally high compared to the others, which is to be expected given the assumptions used to create the product (e.g. Korosov et al., 2018). There are also few years (for example, 2017) when we find persistent substantial difference in the MYI extent comparing the two data products and model.

The rate of decline in MYI extent through winter simulated by the model generally agrees well with that of the CDR data, NSIDC_age data, or both. We also note that when the modelled MYI extent lies within uncertainty of the CDR data early in the winter, it generally remains within this uncertainty until the melt onset. This means that the model is likely to be representing winter processes affecting MYI well. There are a few exceptions to this (for example, 2012) where a steeper decline in the

observations indicates that the model is not losing enough ice during the winter season. This shows that either the assumption we make of FYI always ridging before MYI may not hold for conditions in certain years, or that the model underestimates the overall amount of ridging or export in those years.

In the summer, the CDR data is not available, but MYI extent can be estimated from the weekly ice age data. From 2000 to 2007, the model often overestimates the summer loss of MYI. This may originate from our melting assumption, which holds

better for situations where MYI and FYI thickness are more comparable. After 2007, the model generally matches the observed minima in all years until the summer of 2016, this year being not very well represented in the model in general (Boutin et al., 2023).

In order to evaluate the interannual variability and trend, we consider average January values, shown in Figure 3b. Using the January average eliminates the daily variability prevalent in the CDR data. Using the February and March average values

give qualitatively similar results. Figure 3b shows a clear decline in MYI extent in the model, CDR, and NSIDC_age data. The modelled, CDR and NSIDC_age extents have significant ($p < 0.01$) negative trends of $(-940 \pm 200) \times 10^3$ km$^2$, $(-740 \pm 180) \times 10^3$ km$^2$ and $(-1060 \pm 160) \times 10^3$ km$^2$ per decade respectively. The model, therefore, is consistent with the observed trends, despite the relatively short time series under consideration.

Contrary to the long-term trend, the model struggles to capture the variability of the data. The model and satellite-based

data have a correlation of 0.65 and 0.69 for CDR and NSIDC_age respectively (with $p < 0.01$), but these values drop to 0.24 and 0.12 and are not significant when the data are detrended. This behaviour can primarily be traced to either insufficient replenishment in the model, leading to too little MYI extent for the remainder of winter, or a too-slow decrease in MYI in winter, leading to an overestimate of the MYI extent. These shortcomings generally do not affect the MYI extent and evolution after the summer melt (with the exception of 2016), showing that poor model performance during one winter has limited or no

knock-on effect on the MYI extent in the following years. We note that the two observational datasets do not have the exact same variability, suggesting some uncertainty in the observed sub-annual behaviour.

To demonstrate the evolution of the MYI over time, we show maps of MYI concentration distributions for mid October, January, and April (Figure 4). The years chosen aim to represent different model behaviour compared to the observations, by showing an underestimate by the model (2008–09, panels a–c), an overestimate by the model (2011–12, panels d–f), and a

good agreement between model and observations (2015–16, panel g–i).

In the 2008-09 case (Figure 4a–c), the model captures well the MYI extent extending north from the Canadian Arctic and Greenland in October, and how this MYI shifts southwards and into the Beaufort Sea as the winter evolves. In the Beaufort Sea, an excessively large 'tail' of ice extends westward from the bulk of the MYI pack during the winter. Similar features can be seen in other years (e.g., 2011–12), and here the small features in the observed CDR MYI towards the Chukchi Sea hint at

some MYI existing in this region. The underestimation of modelled MYI extent mostly originates in the Eurasian Basin, where there is an insufficient amount of MYI in the basin from early on in the winter. This points to an insufficient replenishment of MYI in autumn, which the model cannot correct until the following year.

In the 2011–12 case (Figure 4d–f), the model captures the general spatial distribution of MYI in autumn, but with too much ice in the Beaufort and Chukchi Seas. This excess is then maintained throughout winter, as the ice drifts towards the Canadian

coast and into the Beaufort Sea. The overestimate in extent in the Beaufort Sea in October has an amplified effect on the extent in April. The reasons for the persistent positive bias in the distribution for this particular winter is explored further in section 5.1.

In 2015-16 (Figure 4g–i), small differences between the observations and the model in the Beaufort Sea and western Eurasian Basin somewhat compensate each other. More importantly though, the MYI pack displacement towards the coast of the Canadian Arctic Archipelago and Greenland during the winter is well-represented, which serves to preserve the already good initial conditions. Given that the MYI can only undergo dynamical processes and melt once it has been replenished, and melt is negligible from October to April, this analysis suggests that wintertime ice transport/drift within the Arctic Basin is well-captured (which is in line with the drift and winter mass balance evaluation of the model made by Boutin et al., 2023). Discrepancies in spatial distribution, therefore, likely originate primarily from the uncertainty associated with the replenishment of MYI, with uncertainties in ridging probably being of a secondary importance.

The ability of the model to capture the spatial distribution of MYI over the whole simulated time period is summarised in Figure 5. There we compute the sums of grid cells where there is a) MYI in both model and observations, b) no MYI in either model or observations, c) an underestimate of MYI in the model, so that there is MYI in the observations but not the model, and d) an overestimate of MYI, so that there is MYI in the model but not observations. We then convert these to a percent of the Arctic domain (Figure 2c). This is a similar method to that discussed in Aaboe et al. (2019) for their validation of MYI type against ice charts, but here we additionally separate the matching cells into where both ('match MYI') and neither ('match no MYI') find MYI. Figure 5 shows that, on average over the timeseries, the model captures about 80% of the observations in the domain, with a standard deviation of around 5%. The low standard deviation suggests that the portion of the domain that is correctly represented is quite consistent, meaning that the years shown in Figures 4a-c are broadly representative of the model behaviour over the time period.

To further investigate the quality of the results, we look at the spatial agreement in individual sub-regions of the Arctic (Figure 8). We define these regions in a way similar to Boutin et al. (2023) and Ricker et al. (2021). We use four of the outer regions of Boutin et al. (2023) and Ricker et al. (2021), corresponding to the Chukchi, Beaufort, and East Siberian and Laptev seas (the latter two of which are combined as they contain very little MYI). We sub-divide the Central Arctic region of Ricker et al. (2021) into three: the "Central CAA", corresponding to the portion north of the Canadian Arctic Archipelago, the "Central Eurasian", covering the Eurasian Basin portion of the Central Arctic, and the "Central West", covering the western portion. We find that regions of mainly MYI-free or full MYI cover do very well (95% ± 3% for the East Siberian and Laptev seas, 94% ± 8% for the Central Canadian Arctic Archipelago region) while the other regions with more mixed ice cover still perform reasonably (77% ± 13% for the Chukchi region, 70% ± 11% for the Beaufort region, 67% ± 9% for the Central East region, and 76% ± 17% for the Central West region). There are two periods that fall below a 70% match on the pan-Arctic scale: the winter of 2001 to spring 2003, and winter of 2016 to autumn of 2018, which correspond to periods of large uncertainties in the observations (autumn 2001) or too much ice loss in the summer and therefore too little going into the autumn. The performance of the Central East and Central West sub-regions are most affected by these years.

# 4 Processes affecting MYI evolution

## 4.1 General overview of MYI budget

In this section we investigate different processes contributing to the evolution of the MYI area and volume in the Arctic. To this aim, we use the same domain as Boutin et al. (2023), which we subdivide into regions as described in the last paragraph of section 3.

MYI area evolution in the whole Arctic domain is very similar to the one in the domain used to compare extent with observations in section 3 (Figure 6a). MYI area declines over 2000–2018, with a total net loss of $\simeq 1000 \times 10^3$ km$^2$ , which represents over one third of the total Arctic MYI area in 2000. MYI volume also declines. The losses exceed $\simeq 4000$ km$^3$, representing over half of the Arctic MYI volume in 2000.

We now discuss in more detail the contribution of the three sink terms (melt, ridging, and export) and one source term (replenishment) to the MYI evolution over 2000–2018. A region-by-region budget of MYI area and MYI volume, to complement Figure 6, can be found in the Appendix (Figures A1 and A2). Melting dominates the sink terms (Figure 6), accounting on average for 49% of the annual area loss and 75% of the volume loss. Considering the spatial distribution of melting (Figures 7d and 8), we find that the Beaufort region accounts for over 26% of the total MYI area melt on average, despite only covering 14% of the geographic area of the overall Arctic region. The Central CAA region accounts for nearly 20% of the area melt and the Chukchi and Central West regions contribute about 15% each. This key role of the Beaufort Sea for MYI area melt is consistent with observations (Kwok and Cunningham, 2010; Babb et al., 2022). Unlike the MYI area melt, we find that most of the MYI volume melt takes place in the Central CAA region (26%), where most of the volume of MYI is found, while MYI volume melt in the Beaufort region remains large (23%).

Ridging accounts for 24% of MYI area loss on average in the domain, with most of the ridging taking place in the Central Arctic (see Figures 7g and 8). The amount of ridging decreases step-wise (Figure 6), with significantly more ridging as a proportion of the total MYI loss terms before 2008 ($\simeq 30$%) than in 2008 and later ($\simeq 18$%). Ridging is a sink of MYI area but not MYI volume, which means that prior to 2008, around 30% of the total losses in MYI area did not relate to a corresponding loss of MYI volume. In more recent years, ridging accounts for less of the MYI area loss, meaning that more of the losses affecting MYI area can be linked to losses in MYI volume.

Export out of the Arctic domain (of which over 93% is attributable to Fram Strait) accounts for 25% of the volume loss on average, and contributes about 20% of the annual area loss before 2007 and about 32% of it after 2007. This change in MYI area export is because while the absolute MYI area loss due to export remains relatively constant (as for ice export in general, Boutin et al., 2023), the total MYI area decreases, leading to an increased contribution to total MYI area loss in recent years. Within the Arctic domain, some regions experience notable net loss of MYI due to export (such as the Siberian and Laptev seas and the Central West region), while the Beaufort region experiences a large net gain (Figures 8, A1 and A2), the latter providing the source to the large melt that occurs there (Moore et al., 2022).

The sinks of MYI are balanced by replenishment, which is the only source term. In over half of the years, the replenishment does not fully compensate for MYI area and volume loss; replenishment over 2000-2018 amounts to 93% of the total MYI volume losses and 97% of the MYI area losses, resulting in the net losses over the time period.

Looking at the spatial distribution of replenishment (Figure 8), we find that the largest contributor of where FYI is converted to MYI in terms of area is the Central Eurasian region (31%), with the other central regions (West and CAA) also contributing more than the others. However, if we look at the volume, the Central Eurasian is less important (24%), with Central CAA being the most important (29%) as the FYI surviving the summer to be converted to MYI is thinner in the Central Eurasian region than that of the Central CAA region.

## 4.2 Drivers of MYI decline

There is a negative MYI trend for both area and volume. However, there is no trend in any of the source or sink terms that can explain the losses of MYI area and volume; indeed, apart from a reduction in volume replenishment, the other significant trends (ridging and volume melt) show a decrease in contribution over time rather than increase. This is because the actual amount of MYI available to the sink terms also reduces over time. The volume replenishment reduction trend is also not large enough to account for the loss of MYI volume. To account for the decline in MYI area and volume, we recompute trends for the loss/gains as a proportion of the respective MYI area or volume on the 1$^{st}$ January each year. We then find there is no trend in any process. The mean net contributions of the processes and their standard deviations are shown as red diamonds and bars in the left-hand entries of each subplot in Figure 8. If we consider the 2000-2018 mean area or volume change in each region, we see that while on average the change is only slightly negative, the standard deviation is large. Therefore we can expect the long-term changes we see to be driven by anomalous years.

To identify any anomalous years in the timeseries of MYI volume and area, we compute the average net loss in both the full Arctic domain and in the 6 smaller regions for MYI area and volume, as well as the average of each contributing process. We then see if any year falls outside of 1 standard deviation of this average. We do this for the raw values and also for the loss/gains as a proportion of the respective MYI area or volume on the 1$^{st}$ January each year. On a pan-Arctic scale, the MYI areal loss between January 1$^{st}$ on consecutive years reveals two anomalous years: 2007 and 2012. These losses are evident in Figure 6a and Figure 6b (note the "net" losses in red). To put these two years in the context of the overall net losses over the time period, we find that if we remove the total contributions of these years to the budget, the remaining years actually result in a net gain of MYI area. It is notable that these two modelled extreme years coincide with years that set new record low total Arctic sea ice extents (e.g. Parkinson and Comiso, 2013). For MYI volume, two years experience extreme loss, namely 2012 and 2016. Removing these years results in a net gain of MYI volume. Therefore, the negative MYI area and volume trends are associated with an episodic imbalance between the different loss terms and replenishment rather than a constant net loss. To better understand the drivers of MYI loss, we therefore focus on 2007 and 2012, excluding 2016 since the agreement of MYI extent between model and observations is less good in that year.

### 4.2.1 2007

In 2007 (middle bars of Figure 8), we see that ridging is significantly larger than the average (left-hand bars of Figure 8) in the
Central West and Central CAA regions - in Central CAA, it is over 2.5 times larger than the average (Figure 7h and regions
E and C of Figure 8), with small increases in ridging in the neighbouring Central West and Central Eurasian regions as well.
Around 44% the MYI area loss in 2007 is due to ridging (Figure 6b) and almost half (47%) of the ridging occurs in the Central
CAA region (region C of Figure 8). This results in a small increase in MYI area in the Central CAA but a 50% increase in MYI
volume over the course of 2007. There is a net MYI area loss at the pan-Arctic scale but net MYI volume increase (Figure
6a,c), highlighting how the area and volume can become decoupled on both a regional and pan-Arctic scale. Ridging in the
Central CAA is also accompanied by high net areal import into the region (over 8 times the average, which is usually small),
also seen to a lesser extent in the volume import (region C of Figure 8).

Even though most of the ridging and flux anomaly is found in the Central CAA region, we also note that all but the Beaufort
region experiences anomalous export (Figure 8). The Central Eurasian region has almost twice the average export (72% more
than average), despite the Fram Strait, which is usually the main export from both this region and the Arctic Basin, not being
notably larger in that year (Figure 6). As a result of these dynamical changes, replenishment is higher than average (Figure 8),
particularly in the Central West region (Figure 7).

### 4.2.2 2012

In 2012, we see a very different pattern from 2007, with very little ridging, reduced replenishment, and greatly enhanced melt
(Figures 7c,f,i and 8, right-hand bars) compared to the average (Figures 7a,d,g and 8, left-hand bars). The enhanced melt is
most noticeable in the Chukchi and East Siberian and Laptev seas (regions A and D of Figure 8), but all the regions in Figure 8,
except the Beaufort and Central Eurasian regions, experience higher than average melt in 2012. The Chukchi and Siberian and
Laptev seas experience over 65% more areal melt than on average (and 64% more volume melt in the Chukchi region), while
the more central ones (Central CAA and Central West) are more moderate, at 27% and 22% more than average respectively
(and 15% and 17% for volume melt).

Turning to the replenishment, we see that all regions experience less replenishment than average, due to higher than average
FYI melt. There are large reductions in replenishment in the Chukchi (45% less area, 73% less volume than average), Siberian
and Laptev (81% less area, 86% less volume than average), Central Eurasian (63% less area, 47% less volume than average),
and Central West (57% less area, 37% less volume than average) regions. The Central CAA region experiences 57% less areal
replenishment than average, but a small gain in volume. As a result, replenishment compensates 42% (53%) of pan-Arctic area
(volume) losses in 2012, compared to 97% (93%) over 2000-2018. This reduction in FYI available for replenishing is consistent
with the reduced survival rates of FYI in marginal seas feeding the Transpolar Drift in recent years found by Krumpen et al.
(2019).

## 5 Discussion

 ### 5.1 Anomalous years

Given that the extreme years of 2007 and 2012 are not only years of extremely low MYI extent, but also years of extreme September sea-ice extent minima (Stroeve et al., 2008; Comiso et al., 2008; Parkinson and Comiso, 2013), they are of interest for both the evolution of MYI and the evolution of Arctic sea ice in general. Our model results and analysis can therefore bring new insights into to some of the extensive research already conducted to analyse the extreme sea-ice extent minima observed in 2007 and 2012.

One of the main causes of the 2007 sea ice minimum was the compaction of the ice cover towards the Greenland and Canadian coasts due to persistent anomalous winds and a thin ice cover (Zhang et al., 2008; Lindsay et al., 2009; Kauker et al., 2009). The modelling study of Kauker et al. (2009) points to May and June winds being particularly important in this respect. At the same time, Kwok and Cunningham (2012) show that the ice cover continued to converge towards Greenland and Canada for 2.5 months after the minimum. Thus the extreme sea-ice extent minimum in 2007 was caused by both the continued thinning of the ice and an unusual atmospheric state.

Our MYI results for 2007 fit well with the established literature on the 2007 sea-ice extent minimum. The model shows that the loss of MYI area in 2007 was indeed largely due to ridging and compaction of the ice cover, with $830 \times 10^3$ km$^2$ or 44% of the MYI area loss due to ridging. This is by far the largest area lost due to ridging in our simulation (Figure 6), both relatively and in absolute numbers (the average being $350 \pm 180 \times 10^3$ km$^2$, accounting for 24% of yearly losses on average).

The enhanced melt in 2012 has been documented as a big driver of the total sea ice minima in that year, due to an August cyclone occurring on the backdrop of a thinning ice pack (Parkinson and Comiso, 2013; Lukovich et al., 2021) and near-surface ocean heat (Zhang et al., 2013). While the summer melt had a substantial impact on the MYI, we note that reduced replenishment is even more detrimental to the MYI in 2012. The total melt in 2012 is high, 2405 km$^3$ compared to an average of $2143 \pm 582$ km$^3$/year, but the 2012 melt is still short of the maximum amount of melt in our simulation, of 3094 km$^3$ in 2002 (see also Figure 6). Replenishment in 2012 is, however, only 1706 km$^3$, substantially below the average of $2645 \pm 633$ km$^3$/year. In percentage terms, this translates to only 64% of the average volume and 44% of the average area replenishment respectively. It is the second lowest replenishment rate in our simulation, second only to 2017, which we discard as being unrealistic. This low replenishment rate is due to the extensive melt of FYI in 2012, so the 2012 August storm (e.g. Parkinson and Comiso, 2013) should still be considered the main cause of large MYI losses that year. It is, however, important to note that MYI was not lost just because of excessive melt of MYI, but because of excessive melt of FYI which then could not survive the summer to replenish the MYI pack.

Finally, we should note that while the MYI extent modelled in 2012 and preceding and following years compares reasonably well with observations, there is still a clear discrepancy between the two in 2012. Figure 3 shows that while both model, CDR, and NSIDC_age results all agree well on the total MYI extent in autumn 2011, the model clearly underestimates the reduction in MYI extent that occurs during winter. Towards the end of the season (by April 1$^{\text{st}}$) this has resulted in a substantial overestimation of the MYI extent of $704 \times 10^3$ km$^2$. Most of this overestimation is then recovered during the melt season, and

the summer MYI extent agrees reasonably well with that deduced from the NSIDC_age data. Following this, the modelled replenishment is clearly underestimated, resulting in an underestimation of the MYI extent at the end of the melt season by $377 \times 10^3$ km$^2$.

The model underestimation of MYI reduction in winter can only be due to insufficient ridging or export of MYI in the model. While we have direct observations of neither of these factors, we can still deduce something about the importance of each. The modelled total ice area export through Fram Strait, the main export out of the Arctic, is generally very good (Boutin et al., 2023), and from October 2011 to the end of March 2012 there is only a slight underestimate of $12 \times 10^3$ km$^2$ when compared to the observation-based estimates of Smedsrud et al. (2017). Over those months, the MYI export of $301 \times 10^3$ km$^2$ makes up 43% of the total export. If this 43% is an underestimate, this could be a source of the underestimate of MYI loss in this period. Wang et al. (2022) found that over the winters of 2002-2020 the average ratio of MYI area export to total ice area export was 67%. Even with this higher proportion there would only be an extra loss of $165 \times 10^3$ km$^2$, which is significantly less than the model MYI extent overestimation of over $700 \times 10^3$ km$^2$ at the end of winter. The underestimation of MYI loss in winter is, therefore, mostly down to an underestimation of ridging of MYI. This in turn, can be due to an underestimation of the general convergence of the ice cover or due to incorrect assumptions regarding the ridging of MYI (see section 2.3). It is difficult to assess this without an extensive analysis (such as Kwok and Cunningham, 2012, do), and this is beyond the scope of this study. So while summer melt was clearly very important when it comes to the 2012 September minimum, the role of melt in MYI ice loss in 2012 is overestimated in our model, while the role of ridging is underestimated.

## 5.2 Anomalies in ridging

Looking beyond the two extreme years of 2007 and 2012, it is interesting to note that there is a marked change in behaviour of the ice pack when it comes to ridging after 2007. As mentioned, before 2007 ridging constitutes about 30% of the yearly MYI area loss, but after 2007 this fraction is around 18%. The reason behind this change in behaviour is not immediately clear. It is even contrary to expectations, as it is not unreasonable to assume that thinner ice ridges more easily, and that the contribution of ridging in MYI area loss should increase during the period.

If we consider how ridging of MYI evolves over time in different regions, a slightly more nuanced picture appears. Ridging of MYI mainly takes place in the three central regions: Central CAA, Central Eurasian, and Central West. This is not unexpected, given the well-known main circulation patterns of Arctic sea ice (e.g., Colony and Thorndike, 1984), which generally compresses the ice against Greenland and the eastern part of the Canadian Arctic Archipelago, with a return flow through the Beaufort Gyre circulation. Regional reductions in ridging could be explained by atmospheric variability, with conditions less favourable to ridging, but also by the reduction of MYI area in regions prone to ridging and the distribution of MYI and FYI concentration within these regions. Due to our ridging assumption, the MYI fraction of the overall sea ice pack only ridges once the FYI in a given grid cell has been ridged. i.e., the potential for ridging MYI is more important if MYI is compact (with a concentration close to 100%) compared to grid cells with mixed ice types.

In 2007, anomalously high ridging takes place first and foremost in the Central CAA and Central West regions (Figure A1f and j). After 2007, however, much less ridging takes place in the Central West region, but ridging in the other regions is mostly

unaffected. The reduced ridging of MYI can then be attributed to a reduction in ridging primarily in the Central West region. In this case, there is a strong reduction in both MYI area (Figure A1i) and the average concentration of MYI after 2007 (Figure 9). This is because the band of MYI that is compacted and ridged against the Greenland and Canadian coast no longer extends far into the Central West region, meaning that there is much less MYI in that region that can be subjected to ridging. No particular change is observed for the other Central Arctic regions between the years on either side of the 2007 ridging event, either in MYI area or average concentration (Figure 9). The amount of ridging taking place in numerical models remains a challenging quantity to evaluate and an important source of uncertainty (e.g. Hunke et al., 2020). Our assumption that MYI only ridges after all FYI area has disappeared may have affected the importance of the MYI compactness in the total amount of MYI ridging, hence amplified the reduction between the difference pre- and post-2007.

## 5.3   Definition of MYI volume and effects on interpretation

In our implementation of the MYI tracer, we have followed a very literal interpretation of what MYI volume should consist of, namely the volume of ice that has survived one or more summer melts. In particular, this means that basal growth is always classified as FYI, regardless of whether it grows under existing MYI or FYI. Another approach is to define MYI volume as the volume of ice that has the remote sensing signature of MYI, i.e., basal growth under MYI is defined as MYI, similar to how Hunke and Bitz (2009) treat their ice age tracer. As noted by Hunke and Bitz (2009), this approach lends itself well to comparison against observations (e.g., Maslanik et al., 2007; Kwok, 2018; Ricker et al., 2018), where it is only possible to measure the area of MYI and the total thickness of the ice - not the actual thickness or volume of ice that survived the last summer melt.

The two approaches result in substantially different estimates of the MYI volume, as already pointed out by Hunke and Bitz (2009). This difference is shown clearly in Figure 10, which shows a time series of MYI volume using the two approaches (only MYI with a concentration > 0.40 is considered here, as it is our threshold for identifying MYI as an ice type in the simulation). The difference becomes larger during the course of the winter; in our model, we find 74% more MYI volume at the end of March on average if we consider as MYI volume the volume of ice which has the surface signature of MYI, compared to the volume of ice that survived the last summer melt.

These two approaches to estimating MYI volume both have their uses. Using the full ice thickness is clearly better suited for comparison with observations and should be used when doing such comparisons. The approach we use here more accurately traces how much ice survives the summer melt and therefore potentially gives a better idea of how susceptible the Arctic sea ice is to change due to a warming climate. In that context we would like to point out that the loss in MYI volume is less drastic when that is defined as the actual volume of MYI (with an end-of-March trend of -168 $\pm$ 37 km$^3$/year), rather than the volume of ice that has the surface characteristics of MYI (end-of-March trend of -285 $\pm$ 53 km$^3$/year). Our estimate of Fram Strait export of MYI volume is also substantially different, depending on the method used. Using our definition, we find that on average 41% of the export volume is MYI, but using the full ice column thickness we find that 55% of the export volume is MYI.

## 6  Summary and conclusions

We have implemented a novel way of tracing MYI area and volume in a coupled ice-ocean model. By utilising the Lagrangian framework of the sea ice model, we can track when each grid element experiences the end of summer and therefore upgrade the MYI based on physical conditions. This gives a more accurate estimate of the amount of FYI that replenishes the MYI in autumn. The model also allows us to track how the source and sink terms for MYI impact its evolution. The model generally agrees well with multiple datasets, both in integrated amount of MYI extent and its spatial distribution, and the spatial distribution of MYI is good, as long as the minimum ice extent in the autumn is reasonably well captured.

The main drawback of our approach is that as MYI is treated as a tracer, we must make simple assumptions about how MYI and FYI behave when melting and ridging. A possible future approach to avoiding making too simplistic assumptions is to model MYI and FYI as explicit ice classes, rather than as tracers. This would add complexity to the model, but we expect the assumptions needed for that approach to have a better grounding in our physical understanding of the system than the ones made currently.

The MYI cover is affected by three sink terms: melting, export, and ridging, and one source term: replenishment. Melting is the largest sink term, contributing to 49% of area loss and 75% of volume loss on average, and export (almost entirely through Fram Strait) contributing to 25% of the volume loss and 27% of the area loss on average. Ridging only contributes to area loss, at 24% on average. Regional variations reflect the general drift of the ice, with export and import from regions occurring along the Transpolar Drift and into the Beaufort Sea along the Canadian coast. Ridging is most important in the Central CAA region, but also in the Central West and Central Eurasian regions (see Figure 8). Melting occurs in all regions, but is proportionally most important in the Beaufort region. No one process by itself can explain the reduction in MYI area and volume over time.

As the processes we identify affect MYI evolution differently in different years, we selected the two years of extreme MYI reduction to analyse specifically: 2007 and 2012. In the case of 2007, significant redistribution of ice within the Arctic Basin results in a large amount of ridged MYI, most profound in the region north of the Canadian Arctic Archipelago, appearing as a decline in MYI area but not in volume. In the case of anomalous losses in 2012, both MYI area and volume decline on the pan-Arctic scale. The losses are partly attributable to anomalous melting of MYI, but also due to melting of FYI resulting in very little ice left to be replenished in most of the shelf seas.

We also saw a change in behaviour of the MYI following the 2007 minimum. Before the minimum, ridging accounted for about 30% of the area loss of MYI, but after it only accounted for 18%. This means that while there's a strong link between MYI area and volume both before and after 2007, the ratio of the two is not the same before and after. It also suggests that while the finding of Kwok (2018) that MYI area anomalies are closely linked to total ice volume anomalies holds in most cases, it is not always possible to infer the behaviour of MYI volume from MYI area. This is an important consideration when trying to understand MYI volume from area, for example from satellite products, and highlights the use of combining satellite data with models such as this to gain more understanding of what is observed.

Finally, we see that as replenishment is the only source term of MYI (according to our definition), where and when that occurs is important for the annual cycle of MYI. The melt in 2012 is a good example of how MYI area and volume were

reduced, not through the melt of MYI, but through that of FYI, which then did not survive to replenish the MYI cover. This too underlines the point that melt, export, ridging, and replenishment all play a role in the maintenance and decline of MYI and all four need to be considered to gain understanding of its development.

*Data availability.* The ice type product Climate Data Record (version 1) from the Copernicus Climate Change Service (C3S) Climate Data Store (CDS) (2020) is available at https://thredds.met.no/thredds/c3s/c3s.html (last visited June 2020). The ice age product from the National Snow and Ice Data Center (Tschudi et al., 2019) is available at https://nsidc.org/data/nsidc-0611/versions/4 (last visited September 2021). Monthly model outputs for variables discussed in this study are available on Zenodo (https://doi.org/10.5281/zenodo.7785918, Regan et al. (2023)).

## Appendix A:  MYI budget in each sub-region

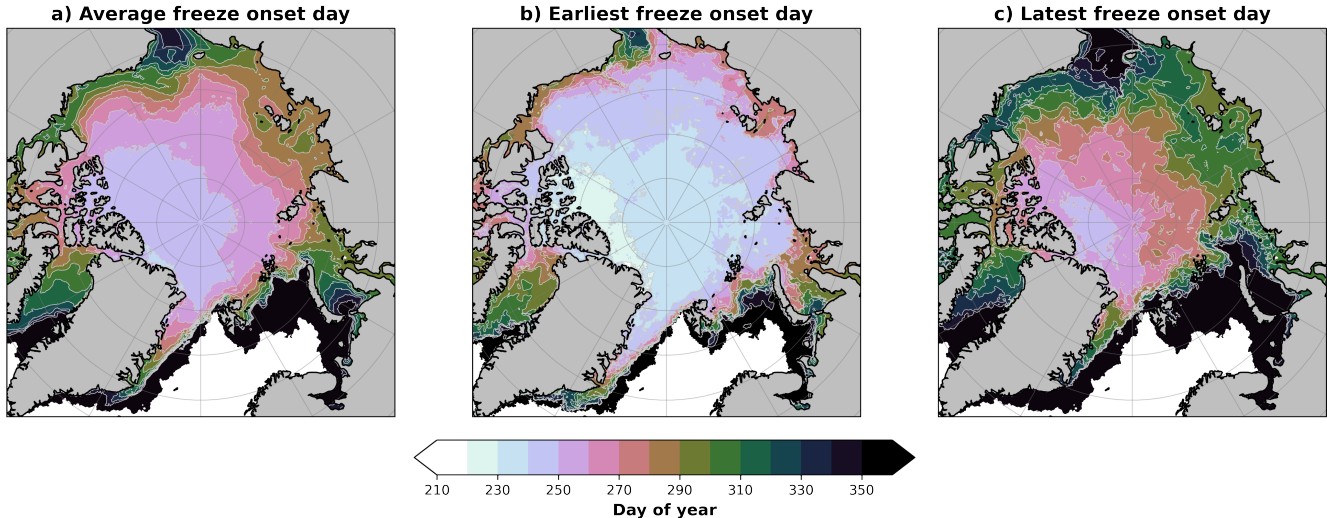

**Figure 1.** Maps showing model results for a) the average freeze onset day over 2000-2018, b) the earliest freeze onset day occurring during 2000-2018, and c) the latest freeze onset day occurring during 2000-2018. The freeze onset day is defined as the first date after 1[st] August when three consecutive days of freezing have occurred, and thus when MYI is assigned for a given year in a given element of the model's mesh.

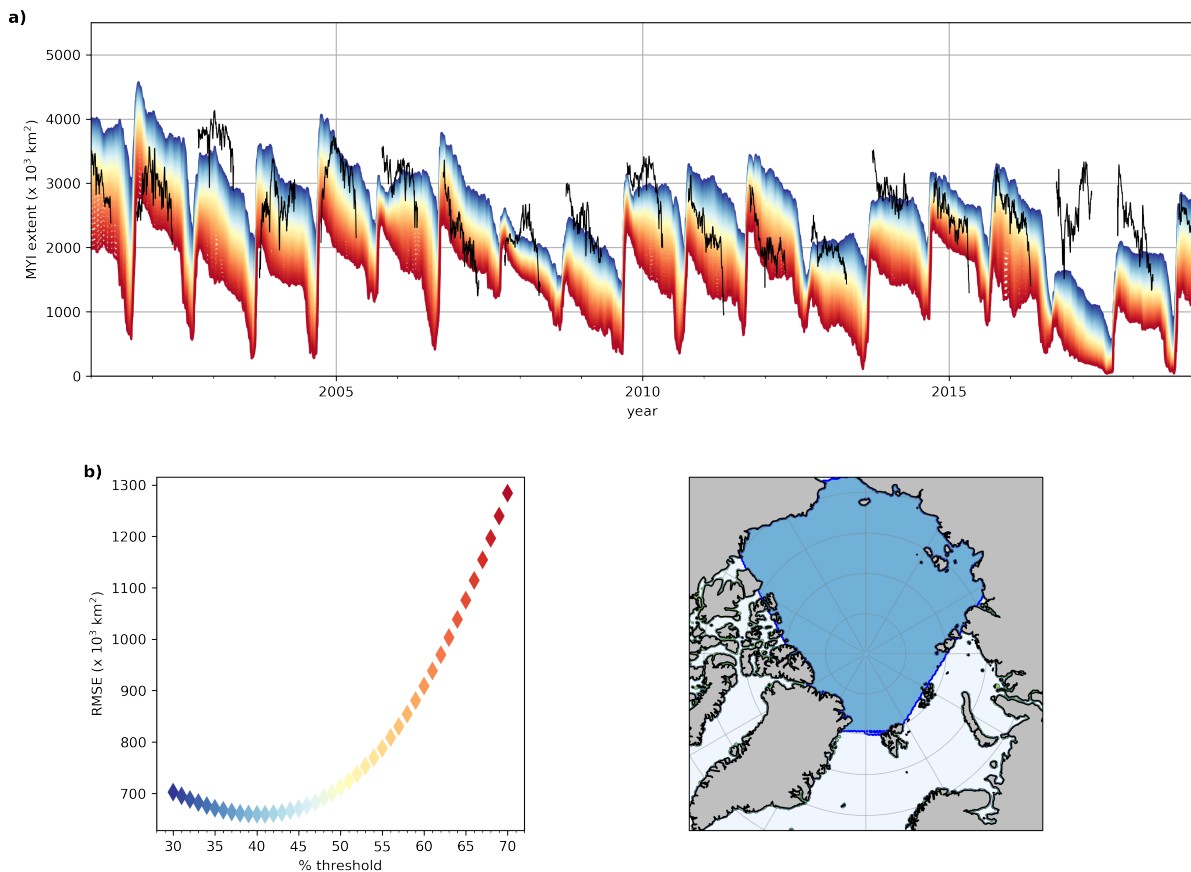

**Figure 2.** a) Timeseries of MYI extent in the Arctic region (blue area in c)). Black lines show the extent based on the OSI-SAF ice type data. Colours show the modelled MYI extent from applying different ice concentration thresholds, ranging from a concentration of 0.30 to 0.70 at 0.01 intervals, as indicated on on the x-axis of b). The daily observed and modelled MYI extents are compared from November to March, and the resulting root mean squared error between the modelled extent for each threshold and the observed extent is shown in b); its minimum is at 0.40. c) The Arctic region (blue) used for evaluation in section 3.

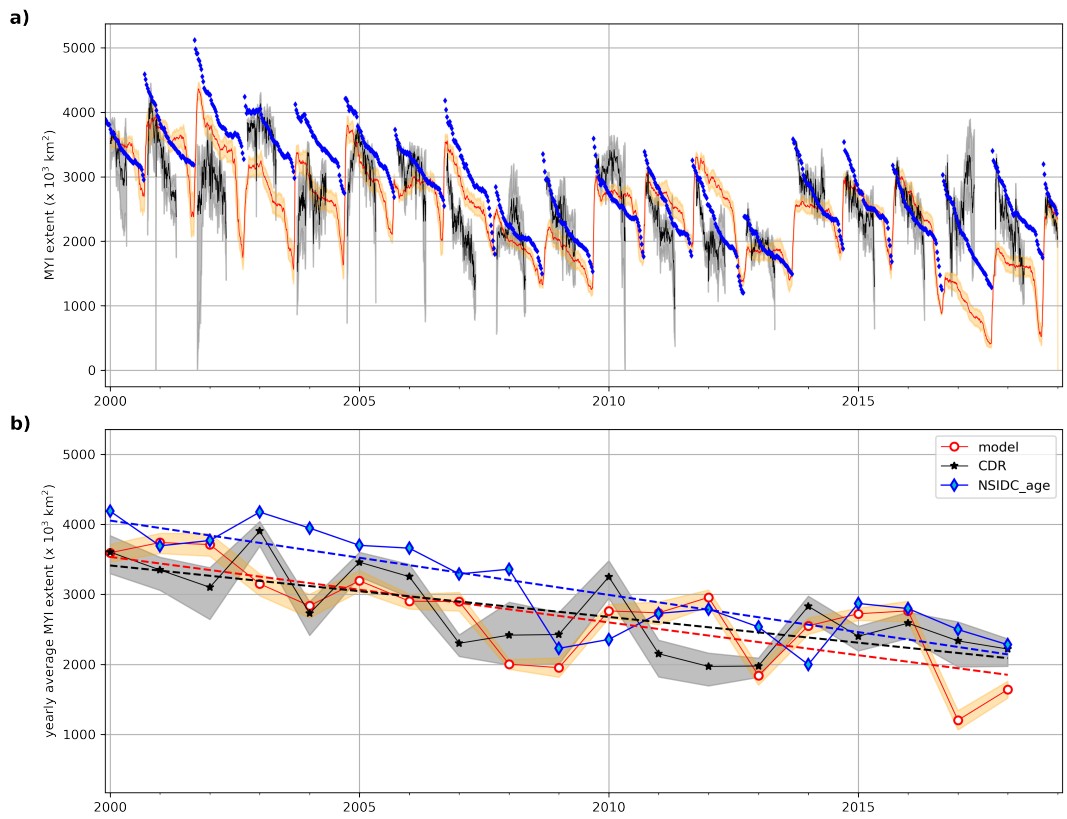

**Figure 3.** Timeseries of MYI extent in the Arctic region (Figure 2c), with the full time series in a) and a time series of the January mean value in each year, along with the associated linear trend, in b). Black lines show the extent based on the CDR data (Aaboe et al., 2019), with grey error bars indicating the range of uncertainty: lower bound includes only those MYI cells with an uncertainty <0.02, while the upper bound includes all MYI cells plus ambiguous cells. The red line shows the modelled extent when using the optimal threshold of 0.40 concentration (Figure 2b) to assign a grid cell as MYI, with the orange band showing the range of extent between a threshold of $\pm\,5\%$. Blue dots show the weekly MYI computed from the age data from NSIDC (Tschudi et al., 2019).

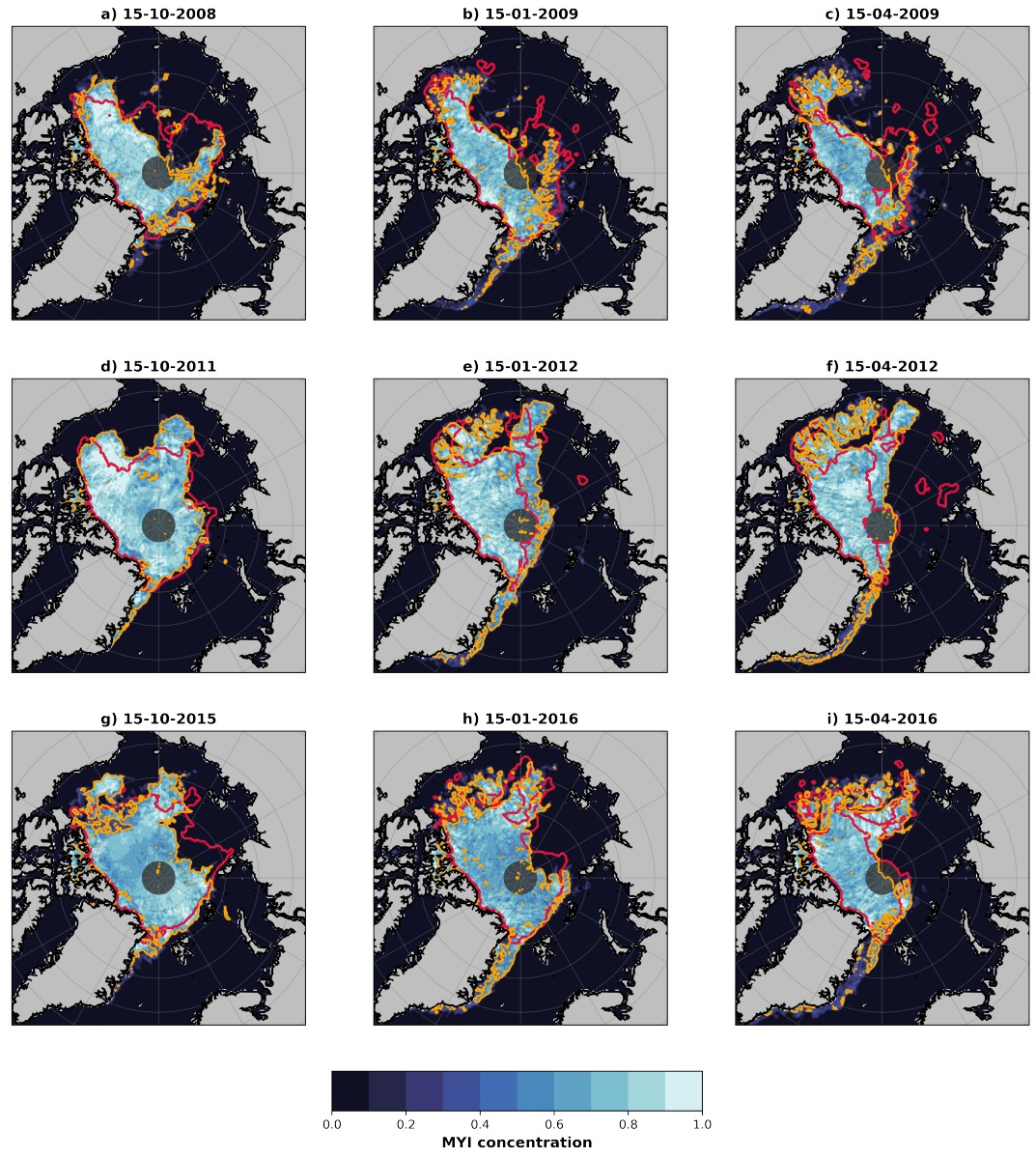

**Figure 4.** Maps of the modelled MYI concentration (shaded), with the modelled extent (concentration of 0.40) shown in orange, and the CDR extent shown in red. Columns indicate different stages, specifically the 15$^{th}$ October, the 15$^{th}$ January, and the 15$^{th}$ April. Rows indicate winters of 2008-2009, 2011-2012, and 2015-2016, respectively. Grey shaded region north of 88 $^{\circ}$N shows where CDR data is missing.

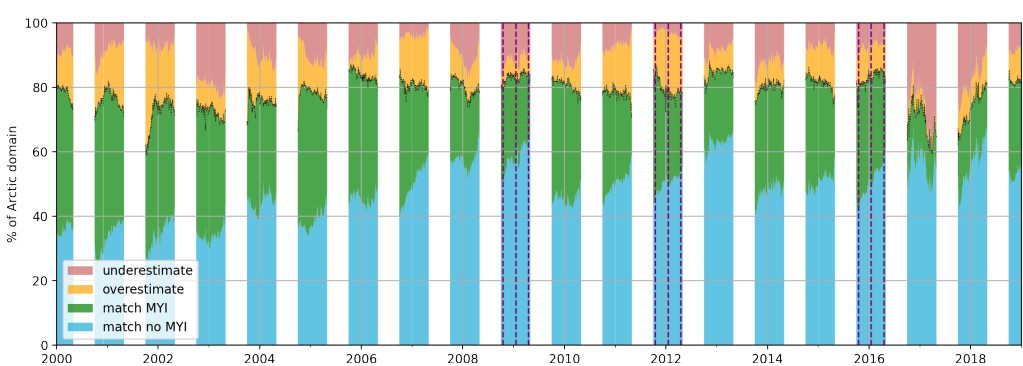

**Figure 5.** Agreement of the model results with the CDR MYI over 2000-2018. Colours show the percent of the Arctic region (Figure 2c) that the modelled extent and CDR data agree that cells are either MYI (green) or not (blue), the percent where the model finds MYI but the CDR does not (yellow), and the percent where the CDR data has MYI but the model does not (pink). Black horizontal dotted line separates regions of agreement (below) and disagreement (above). Purple vertical lines indicate the relative percentages on the dates of the maps in Figure 4.

.

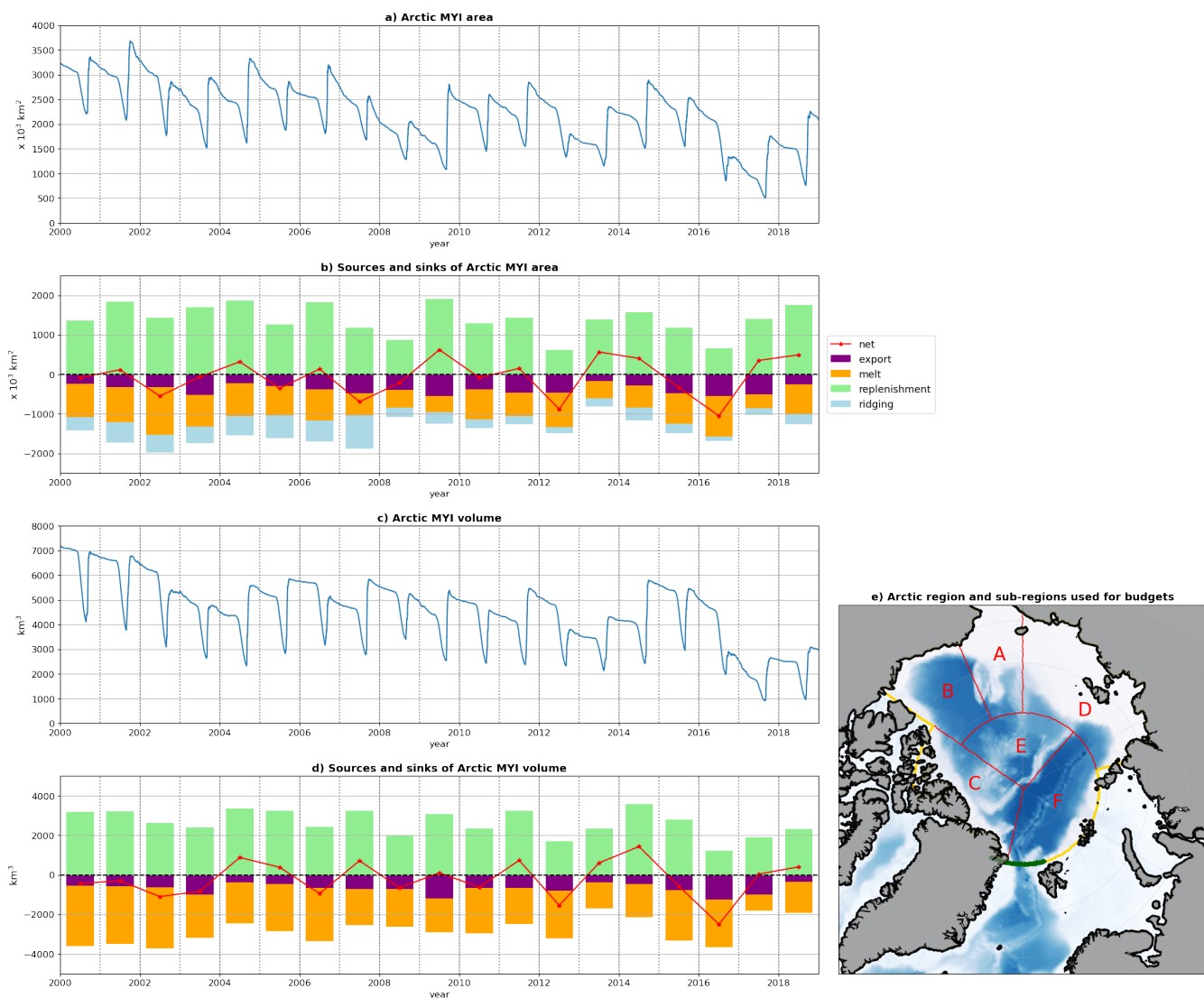

**Figure 6.** a) Timeseries of MYI area in the Arctic region (subplot e)). b) Yearly total contributions to the MYI area, as follows: (purple) loss due to export from the Arctic region (predominantly through Fram Strait), (orange) loss due to melt, (light blue) loss due to ridging, and (green) gain due to ice that has survived the summer in addition to the MYI already present. The net contribution for each year is shown in red. Subplots c) and d) show the same fields but for the MYI volume; note that there is no ridging contribution to the volume budgets as it is a conservative process for this quantity. e) The Arctic region (yellow outline) used for the budgets in a)-d), along with sub-regions (red) used in further analysis: A = Chukchi, B = Beaufort, C = Central Canadian Arctic Archipelago, D = East Siberian and Laptev, E = Central West, and F = Central Eurasian. The sub-regions can also be found on Figure 8. Section used for Fram Strait export is shown in dark green.

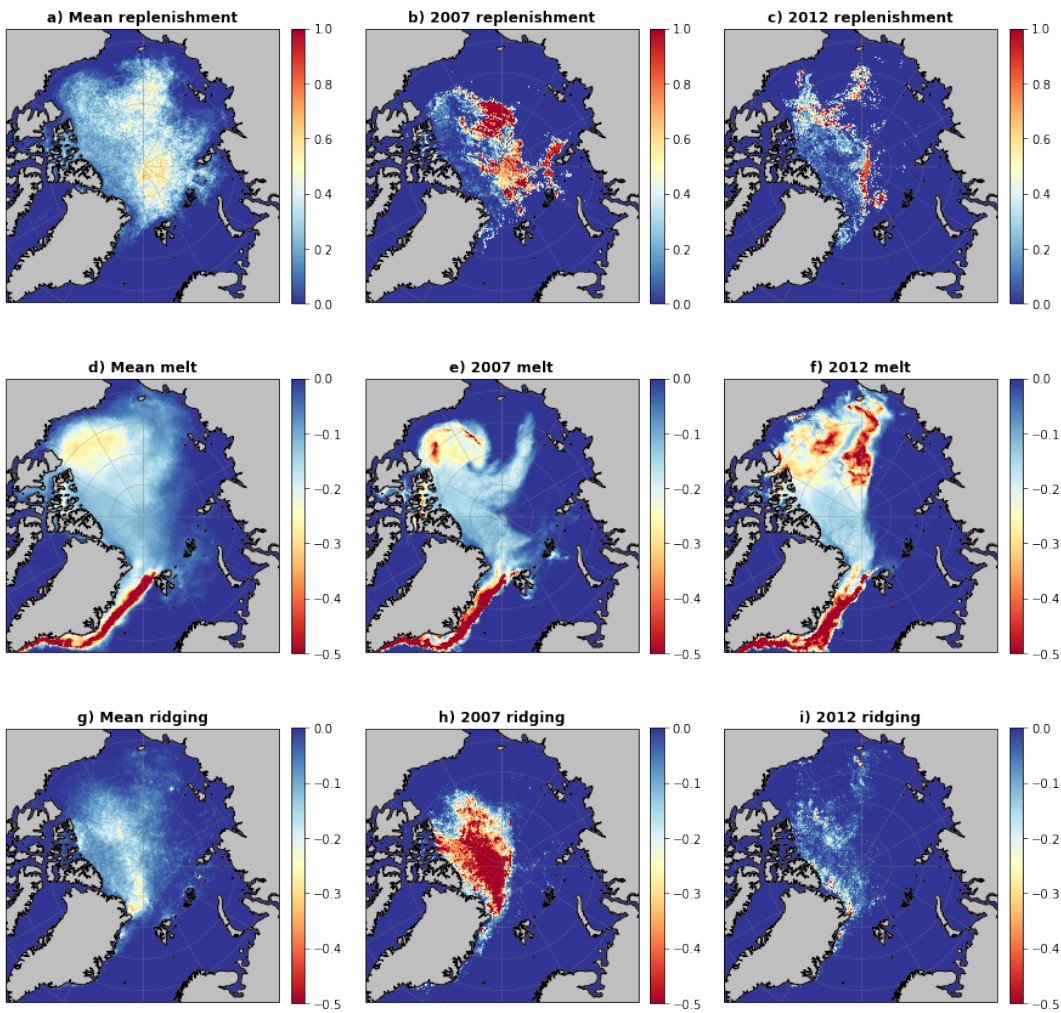

**Figure 7.** Maps showing the 2000-2018 average of the yearly contribution of replenishment (first row), melt (second row) and ridging (third row) to the concentration of MYI (first column). Second and third columns show the total contributions to the years 2007 and 2012 respectively.

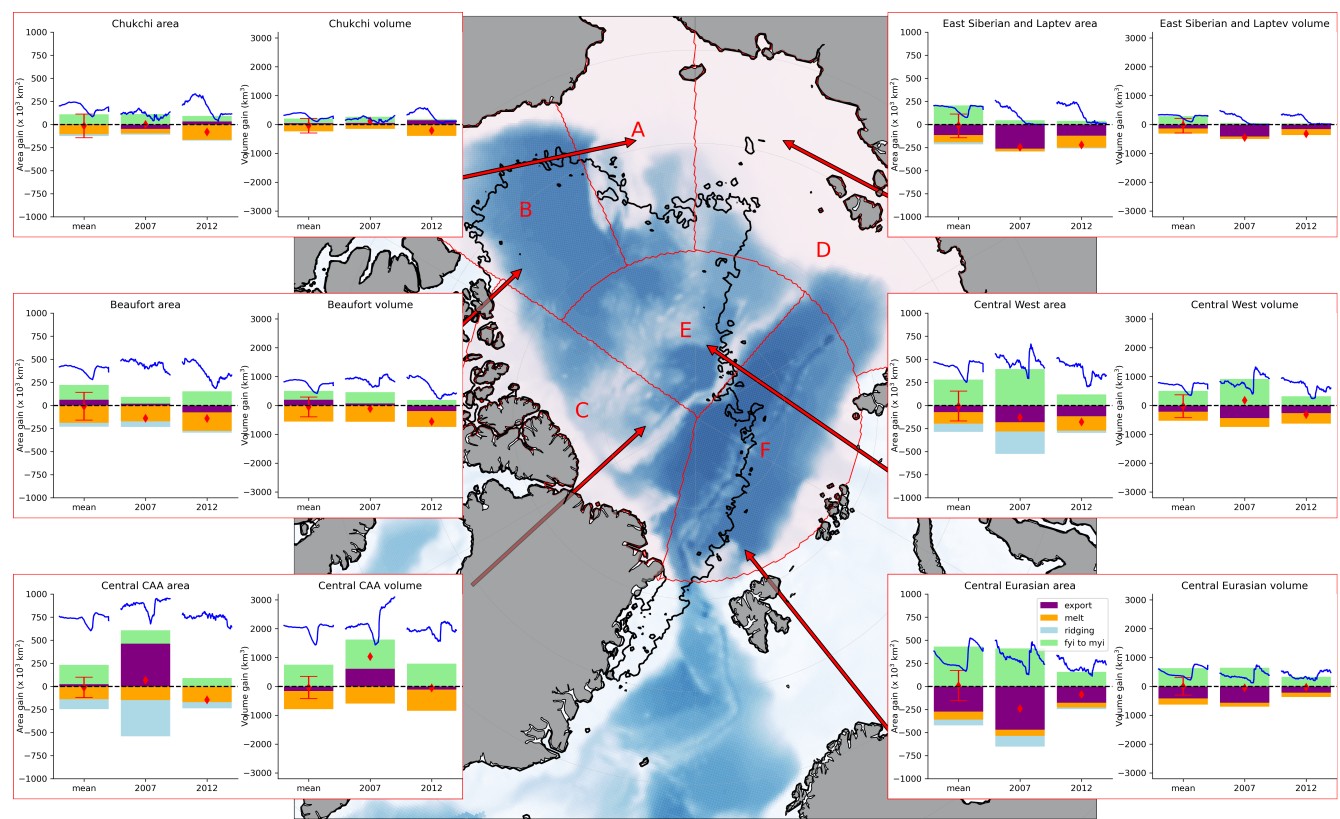

**Figure 8.** MYI area and volume budgets for individual regions: the Chukchi Sea (A; top left), Beaufort Sea (B; middle left), Central Canadian Arctic Archipelago (C; bottom left), East Siberian and Laptev seas (D; top right), Central West (E; middle right), and the Central Eurasian (F; bottom right). Bars are shown for the 2000-2018 average (left bar), and the contributions of each process in 2007 (middle bar) and 2012 (right bar). Net contributions are shown by red diamonds, with error bars on the average indicating one standard deviation of the yearly net values from the average. The mean, 2007 and 2012 seasonal cycles of MYI area and volume in each region are shown in blue.

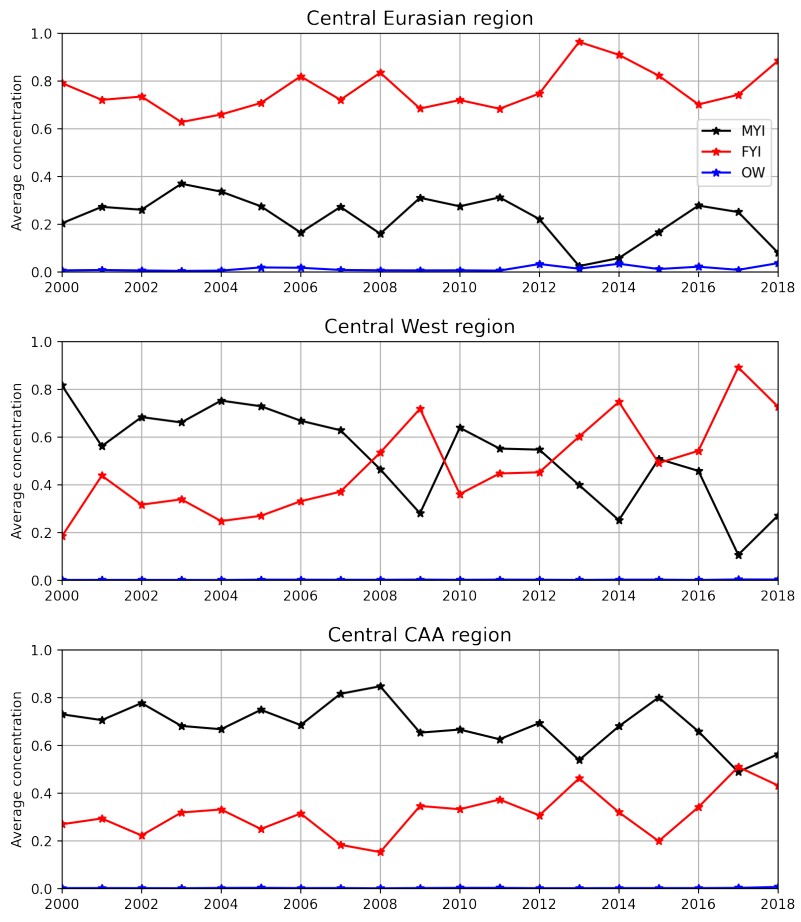

**Figure 9.** Timeseries of average February concentration of MYI (black), FYI (red), and open water (OW; blue) in each of the central regions: Central Eurasian (top), Central West (middle) and Central CAA (bottom).

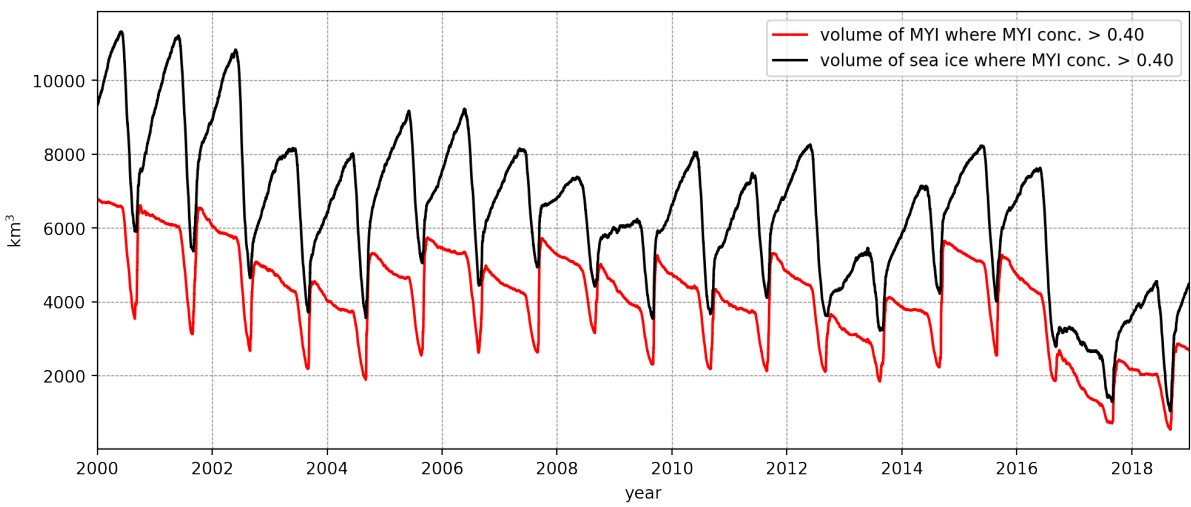

**Figure 10.** Timeseries of volume of MYI (red line) contained within the MYI concentration contour of 0.40 (the extent threshold which best compares to satellites determining ice type). For comparison, the total volume of sea ice within the MYI concentration contour of 0.40 is also shown (black), which is more comparable to the method used by Kwok (2018).

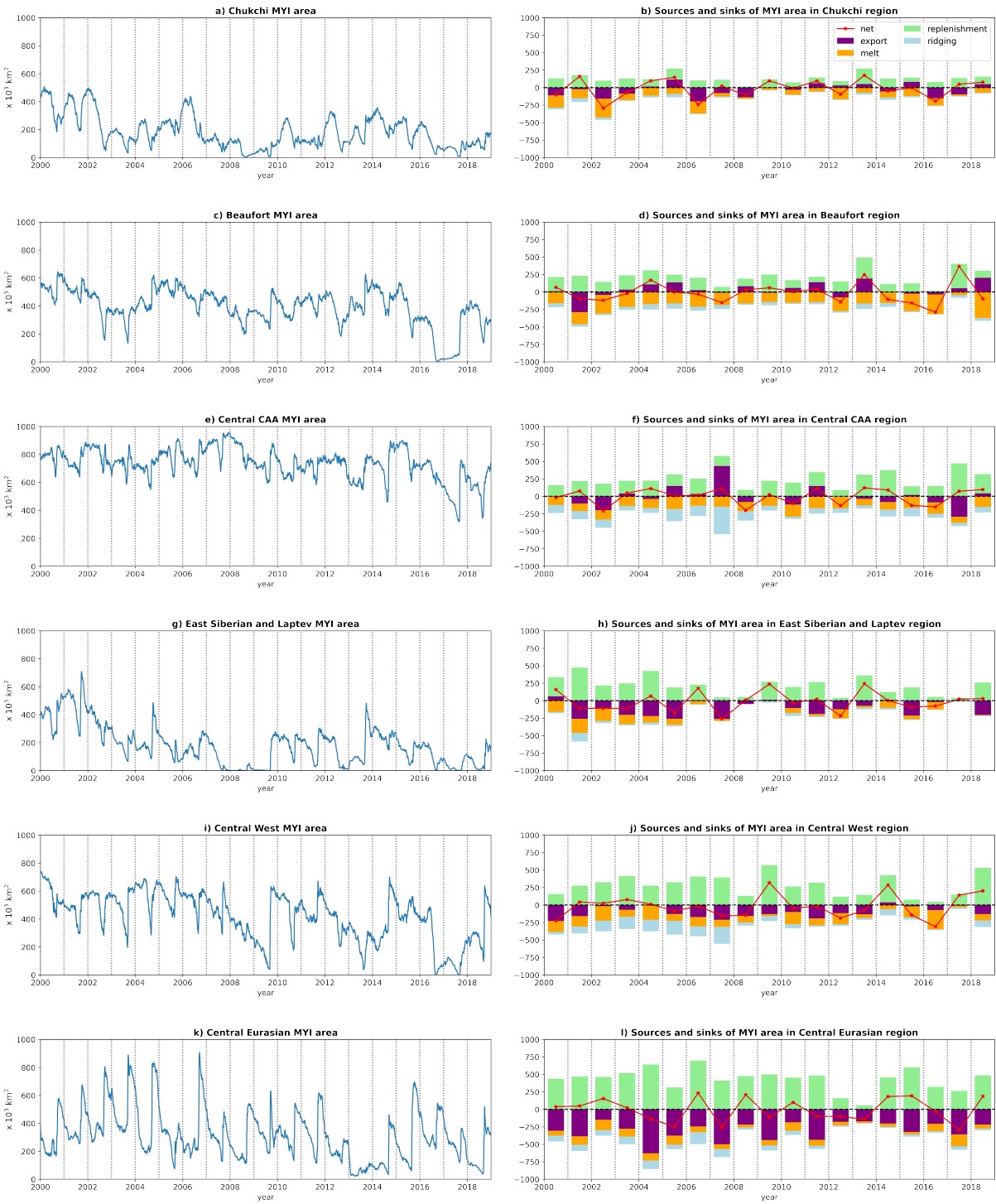

**Figure A1.** MYI area (left column) and source and sink terms (right column) for each of the sub-regions defined in Figure 8.

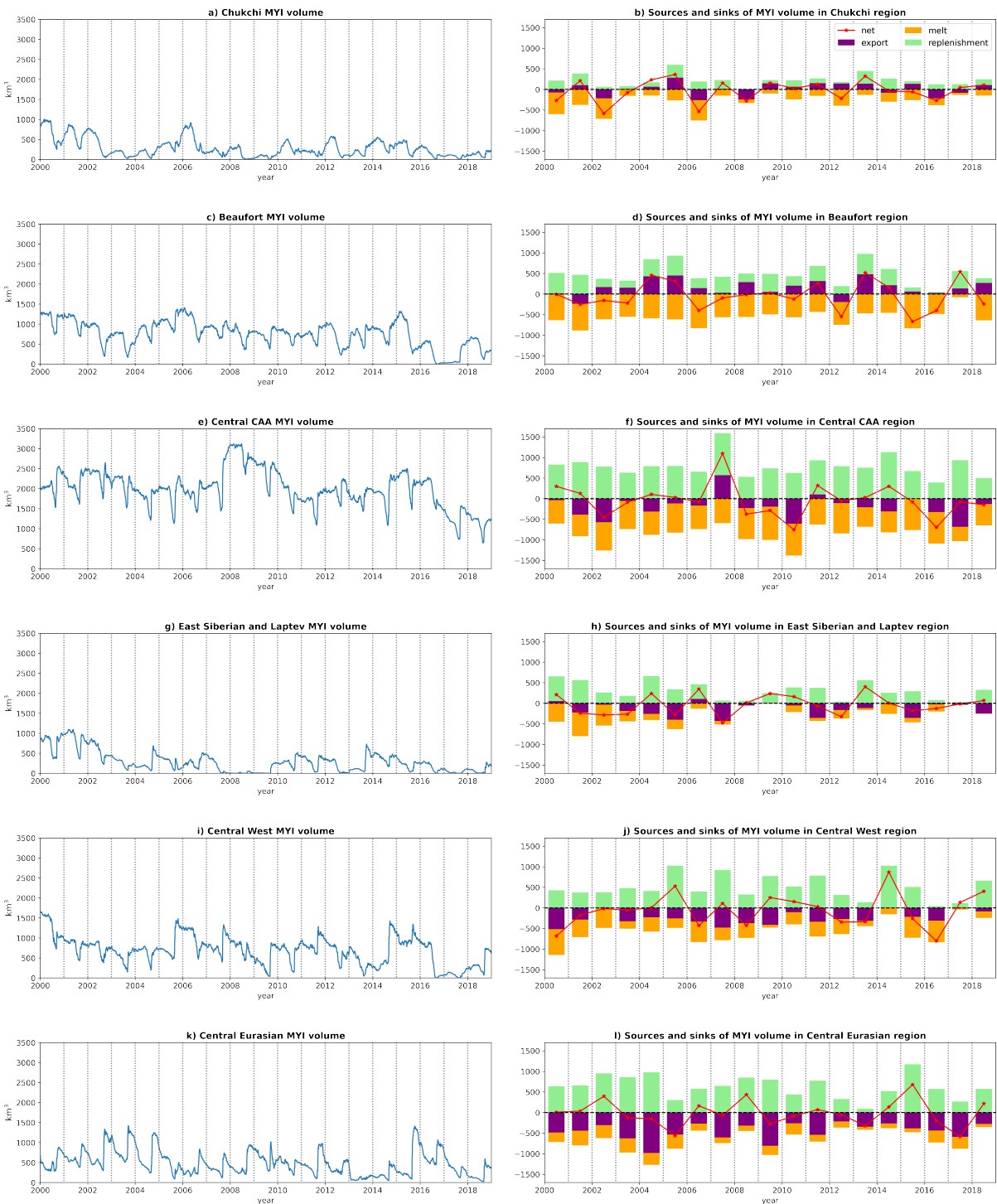

**Figure A2.** MYI volume (left column) and source and sink terms (right column) for each of the sub-regions defined in Figure 8.

*Author contributions.* PR and EO obtained the funding. PR, EO and HR formulated the study. HR, EO and GB developed the new code for tracking MYI. HR carried out the analysis. GB produced the simulation and helped with analysis. AK assisted with evaluation against observations. HR wrote the manuscript with input from all authors.

*Competing interests.* The authors declare that there are no competing interests.

*Acknowledgements.* This research has been funded by the Norwegian Research Council (FRASIL: grant no. 263044 and Nansen Legacy: grant no. 27673) and JPI Climate and JPI Oceans (MEDLEY project, under agreement with the Norwegian Research Council, grant no 316730). The computations were performed on resources provided by Sigma2 - the National Infrastructure for High Performance Computing and Data Storage in Norway (project nos. NN9878K and NS9829K). We are very grateful to two anonymous reviewers for their constructive feedback which helped us to improve the manuscript.

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
