# Peer review of "Modelling the evolution of Arctic multiyear sea ice over 2000-2018"

_The Cryosphere, 2022_

## Author Comment (AC1)

We thank the two reviewers for their positive and constructive comments and suggestions, and have taken them on board. The original comments from each reviewer are in bold, with our response below each in regular font and the associated line numbers from the new, updated manuscript in bold. We have attached both the new manuscript, with the changes incorporated, along with a version highlighting the changes from the original submission and the new manuscript. At the end of the review

We also note that the red net contribution from the processes in the volume budget of Figure 6(d) had duplicated the net line from the area (6b); we have fixed this now.

As the tracked changes does not highlight the new references added, we have listed the additions to the new manuscript at the end of our response.

Review 1:

**The manuscript "Modeling the evolution of Arctic MYI cover over 2000 - 2018" by Heather Regan uses a sea ice-ocean model to examine how MYI area and volume change in the Arctic. The advantage of the presented approach is that, compared to other studies based on satellite data, budgeting of different sink and source terms such as ridging, melt and replenishment can be done and linked to the MYI retreat. The authors conclude that rather than one process dominating the observed losses. Furthermore, they take a closer look at the processes controlling MYI area and volume changes in anomalous years like in 2007 and 2012.**

**The study is definitely worth publishing. It is a great addition to existing (remote sensing) studies, excellently written and very well and clearly structured. The model validation and comparison with observations is extensive and convincing. The budgeting of the individual processes contributing to the decline of MYI in the Arctic is comprehensible. The focus on the two extreme years in 2007 and 2012 is also reasonable.**

**I believe that the paper will be widely cited and can make an important contribution to a better understanding of the changes in the last ice areas.**

We thank the reviewer for their review and really appreciate their positive comments and feedback on the manuscript.

**My comments are mainly "minor". I do, however, encourage a somewhat deeper investigation or at least better explanation of the post-2007 decline in ridging (see my comments below).**

**If possible, please provide a Figure like Fig. 6 for the individual key regions (i.e. in the appendix).**

We understand the point about further detailing the post-2007 decline in ridging and have provided an explanation of how we have addressed this in response to the relevant comments below. In particular, we have added a new figure to help demonstrate the causes of the decline (new Figure 9). We see the interest in providing budgets for the individual regions and have added this to the appendix. We refer to this figure to help explain the cause of the abrupt decrease in ridging. We hope that is now more clear for the reviewer and other readers.

"A region-by-region budget of MYI area and MYI volume, to complement Figure 6, can be found in the Appendix (Figures A1 and A2)." **(LINES 298-299; Figures A1 and A2)**

**Comments:**

**Line 24: Isn't the statement that "MYI area anomalies being closely linked to anoamlies in Arctic ice volume" contradicting to what is said in the abstract? Line 11: "…can result in large reduction of MYI area without a corresponding loss of MYI volume"? Maybe you can emphasize this contradiction in the abstract since it points to the importance of this study.**

The phrasing of this might imply that we contradict that study, but actually we do find that in most cases this agreement holds well. The paper in question (Kwok, 2018) finds that, on a subset of the time period in question, the R2 value of MYI area anomalies against total ice volume is 0.85. Total ice volume anomalies closely follow MYI volume anomalies in our model. If we carry out the same analysis as Kwok (2018), both against total ice volume and MYI volume, we actually obtain the same R2 value as them, although our variability is slightly different. What we would like to emphasise is that in some situations this relationship does break down (for example, in 2007 in our model), and therefore the MYI volume/total ice volume cannot always be inferred from the MYI area, due to dynamic processes affecting area and volume in different ways. We have added a line in the conclusion to refer back to (Kwok, 2018) and clarify our statement. We also added "in years such as 2007" to the abstract to emphasise that 2007 was an extreme year that led to the breakdown in the relationship.

"It also suggests that while the finding of Kwok (2018) that MYI area anomalies are closely linked to total ice volume anomalies holds in most cases, it is not always possible to infer the behaviour of MYI volume from MYI area." **(Lines 512-514)**

**Line 43: "ice type classification fails in summer": I don't think this can be generalized, because of course different methods and sensors are used. Some of them are more robust in summer.**

This is a good point.

We have edited this to say "ice type classification is difficult in summer, due to surface melt (e.g. Kwok, 2004; Aaboe et al., 2019) and therefore some products limit availability to just the winter months." **(Lines 43-45)**

**Line 46: A little bit outdated reference: There are more recent studies on reliability of motion products in summer available. Also the products get better and better. E.g. Hiroshi Sumata (Tromsoe) did quite some work on product intercomparison.**

We have added the reference to Sumata et al (2014) and also rephrased slightly to note that improvements are being made: "...which are improving but have historically been significantly less reliable in summer" **(Lines 46-47)**

**Line 50: "…challenging". May be refer to "von Albedyll 2021, Linking sea ice deformation to cie thickness redistribution using HR satellite and airborne obs.**

Thank you for your suggestion - we have added this reference to the text **(Line 51)**

**Line 52: I guess there are more recent publications assessing the accuracy of altimetry missions. E.g. the Nature Paper by Landy (2022). Avoid terms like "relative uncertain"**

We have removed the phrase "and even that is relatively uncertain", and added a reference to the suggested Nature paper: "with year-round information only being available very recently". **(Lines 52-54)**

**Paragraph 42 - 55: In this paragraph, the author focuses strongly on the drawbacks of the satellite-based methods. I think that this is not necessary, because the advantages of the models are obvious. Its just a comment, no need to change anything J**

We apologise if it came across as being overly negative towards satellite observations. This was not our intention; rather, we wanted to emphasise that there was room for models to help with our understanding. The other edits that we have carried out in response to the other comments above hopefully reduce the emphasis on the drawbacks and highlight some of the recent developments more clearly to make the paragraph more balanced. **(Lines 43-55)**

**Line 89/70: MYI**

Thankyou for pointing this out - we have changed this. We also changed other instances where MYI was "multiyear ice", and introduced FYI earlier.

**Line 125: Just out of curiosity: How well do the results on temporal and spatial variability agree with satellite-based methods? Satellite-based methods probably rely on changes in surface properties, while temperature differences and heat fluxes come into play in models? However, in general, model and satellite data on freeze up should be comparable?**

Satellite-based methods that rely on changes to surface properties, such as the OSI-SAF product (which uses brightness temperature) that we compare against in this study, are capturing a surface signature of MYI that has undergone melt during at least one summer season before being upgraded from FYI, and therefore the processes that determine its signature are directly impacted by temperature and heat fluxes. This procedure is what we try to mimic when we identify cells with sea ice at the end of summer. The OSI-SAF product is not available in September so we are unable to directly compare against this product during the freeze-up.

**Line 143: "MYI is both thicker and stronger…": This assumption probably holds one of the largest uncertainties: In particular, in the marignal ice zones and throughout the Transpolar Drift, FYI and SYI (i.e., MYI in this study) are likely to have similar thicknesses and are otherwise difficult to distinguish. When is ridging in the model considered to be completed in an ice age class? Or in other words: At what point has enough FYI been deformed for MYI to proceed?**

We agree this is one of the largest sources of uncertainty. Since neXtSIM uses a moving mesh, convergence/ridging causes the triangles of the mesh to become smaller. When this happens, the model simply assumes that the area of MYI is conserved, as long as this area remains smaller than the area of the triangle. Or in other words, MYI only ridges when all the FYI has been ridged onto the MYI. This acts as a lower bound for MYI ridging, especially in the later years of the simulation when a) there is less difference between the thickness of MYI and FYI, and b) the average MYI concentration in a grid cell decreases, meaning that there is more open water and FYI to converge in before the MYI is reached. As noted by the referee, this also extends to MYI in the MIZ or Transpolar Drift area where the difference in thickness between MYI and FYI is smaller.

We have emphasised this point and clarified our working hypothesis in the text:

Convergence, through ridging, of ice acts as a sink term for area only, not affecting volume. In the model, we assume that MYI ridges only after all FYI is ridged. In practice, convergence causes the triangular elements of the mesh to become smaller. If the element is fully ice covered, then ridging occurs. When this happens, the model simply assumes that FYI area is reduced (if there is any) and the area of MYI is conserved, as long as this area remains smaller than the area of the triangle. If this is not the case, MYI ridges and its area is reduced. This choice is based on our expectation that MYI is generally both thicker and stronger than FYI and so nearly all, if not all, ridging should take place within the FYI area, as long as this exists. This hypothesis is source of uncertainty as it may underestimate MYI ridging in areas where MYI and FYI have fairly similar thickness (like the MIZ or in the Transpolar drift). It may also be less representative in recent years, as the observed MYI thinning is faster than the one of FYI (e.g., Kwok, 2018). **(Lines 155-163)**

**Fig 2: All very interesting!**

We thank the reviewer for this comment!

**Line 191: May be you can get in touch with the producer and ask what the reason may be?**

We have contacted the producer but have not yet had a response. We are certainly interested in trying to pin this down more.

**Line 201: "…conditions in certain years" and areas. See my comment to Line 143: I believe that in the marginal ice zones the assumption may not be correct**

This is likely the case, yes. We have expanded on our description of this assumption in section 2.3 (sources and sinks of multiyear ice) to emphasise this more clearly, as well as expanding on this limitation in section 5.2 (anomalies in ridging). **(Lines 161-163; section 5.2)**

**Line 209: CDR data.Using**

Apologies - we have added a space now

**Line 2014: "The model and satellite-based data?**

Yes - we have clarified this in the text **(Line 234)**

**Line 250: Just a general comment: I find the evaluation and comparison with the CDR and NSIDC data highly interesting and well done. Thanks, I appreciate reading**

We thank the reviewer for this comment!

**Fig. 6e) Hard to read. Can you make it bigger or refer to Fig. 8**

We have tried to make the inset bigger without compromising too much on the rest of the figure. We also now refer to Figure 8 in the caption, and have rearranged the letters so that they read down in two columns, which is more intuitive when relating to the subpanels in Figure 8. **(edited figures 6 and 8)**

**Line 280: I get what you saying and I find it interesting. May be you can clarify statement a bit better**

We have changed to

"Ridging is a sink of MYI area but not MYI volume, which means that prior to 2008, around 30% of the total losses in MYI area did not relate to a corresponding loss of MYI volume. In more recent years, ridging accounts for less of the MYI area loss, meaning that more of the losses affecting MYI area can be linked to losses in MYI volume." **(Lines 310-313)**

We hope it is now clearer.

**Fig 8: Can you add notations of Fig. 6e? (A,B,C,etc)**

Good idea - we have done this. We have also rearranged the letters so that they relate more intuitively to the subpanels.

**Line 291: The CE is the largest contributor because this is where most of the FYI turns into SYI?**

Yes, that's right. With the new budgets in the appendix it can be seen that this is also increasing as it moves away from being more covered in MYI in the past. We have clarified in the text:

"we find that the largest contributor of where FYI is converted to MYI in terms of area is the Central Eurasian region (31%)..." **(Lines 324-325)**

**Line 294/295: I am sceptical about the conclusion. May be move it to the discussion.**

We agree with the reviewer that this is a bit speculative. We have rephrased to remove the suggested reason behind the thickness:

"...as the FYI surviving the summer to be converted to MYI is thinner in the Central Eurasian region than that of the Central CAA region." **(Lines 327-328)**

**Line 296: I guess you refer to Fig. 8 in this statement, but are you sure there is no trend if looking at the Arctic wide sources and sink terms? To me it seems that ridging is reduced, although it has this staircase appearance.**

We apologise for how this is phrased. We did various tests of trends and significance and unfortunately did not summarise it well. You are indeed right that on an Arctic-wide scale, there is a significant trend in some of the contributors to the budget - namely ridging for area, and melt, replenishment and the sum of the sink terms for volume. But these significant trends suggest that the contributions from those terms are reducing over time; so while less replenishment leads to less MYI, for the other terms it would suggest less contribution to the budget over time, not more. This is because, for the sink terms, the amount of MYI area or volume in the domain also reduces over time, and therefore there is less to act as a sink on. We therefore do not believe that the trends on the raw values give us much information on how the processes are changing. We did the trend analysis again on the processes divided by the amount of MYI area or volume on January the 1st (as we also do for the anomalous years in the next paragraph), and this time there are no significant trends in any process. We have added a few sentences to explain this thought process in a concise way, and hope that it is now clearer.

"However, there is no trend in any of the source or sink terms that can explain the losses of MYI area and volume; indeed, apart from a reduction in volume replenishment, the other significant trends (ridging and volume melt) show a decrease in contribution over time rather than increase. This is because the actual amount of MYI available to the sink terms also reduces over time. The volume replenishment reduction trend is also not large enough to account for the loss of MYI volume. To account for the decline in MYI area and volume, we recompute trends for the loss/gains as a proportion of the respective MYI area or volume on the 1$^{st}$ January each year. We then find there is no trend in any process." **(Lines 330-336)**

**Comment to export: How well does the modelled (Fram Strait) export compare with exports from others? Ricker, Smedsrud, etc? Does it capture the seasonal and interannual variability correctly?**

- For total sea ice export: Boutin et al (2023) have compared to the area estimates of Smedsrud et al (good) and volume estimates of Spreen et al (generally good, sometimes underestimated). So yes, variability is good. This study did not compare against Ricker et al (they use different gates), but given that those estimates are larger than Spreen, we would likely underestimate the ice volume transport compared to that.
- We have not compared MYI export estimates. To our knowledge, RIcker et al., 2017 is the only study to partition between FYI export and MYI export. But they use ice type classification at the surface, leading to large percentages of MYI in the volume export. Our original Figure 9 (now Figure 10) demonstrates that our setup, as it is, cannot be directly compared with observations that use this philosophy because we do not include new ice basal growth as MYI and therefore our estimates will be lower by definition.
- We actually had the following text in a draft of the paper in relation to Figure 9: "Note that the figure of 55% is obtained by only including sea ice volume within the MYI contour of 0.40, found to be the optimal contour describing MYI over the full timeseries and the full Arctic. If we use a lower threshold by which to include MYI, for example, 0.20, the proportion of ice export made up of MYI goes up to 74% - similar to the average of 80% observed by Ricker et al (2018) over 2010-2017." We removed it as our focus is more on how MYI export contributes to overall MYI evolution, as opposed to how it contributes to total ice evolution. We did not want to overcomplicate the text with too many percentages of different things, and think that the discussion around the choice of threshold may also complicate matters further,

particularly as this is dependent on an optimal threshold for ice type classification from the Arctic Basin being applicable to the Fram Strait region which it may not be.

We have moved the reference to Boutin et al (2022) to the end of the Model setup subsection and added a few sentences after to describe briefly how Boutin et al (2022) was validated.

"More details of the coupled neXtSIM-OPA model setup and an evaluation of the sea ice properties can be found in Boutin et al (2022). In brief, sea ice extent, volume and drift are found to agree generally well with observations, especially for the drift. Sea ice volume export through Fram Strait - particularly its variability - is also consistent with observed estimates, but may be underestimated prior to 2008 (the observations show large uncertainties). Boutin et al (2022) also compared their dynamic and thermodynamic components of the winter mass balance against estimates by Ricker et al (2021) for the period 2003-2018. The model shows a reasonable match for the thermodynamics, and is able to capture the variability of the dynamic changes in sea ice volume." **(Lines 89-95)**

**Chapter 4.2.1: Great chapter. However, it took me a while to understand it all. Maybe you can refer to Figures more often in one or the other place? Same for 4.2.2**

Apologies that this was unclear. We have added figure references - notably referring to the region letter (A-F) to help signpost the reader to the relevant part of Figure 8, and detailing the relevant bars (left, middle, or right) to look at at the beginning of each subsection. We hope this helps.

**Line 332: I guess the replenishment rate is directly related to the FYI area available at the end of the summer. May be just state that more FYI was melted then usual, such that replenishment rate was reduced? I hope I did not get this wrong though. Note that according to this study (https://doi.org/10.1038/s41598-019-41456-y) there is a generally reduced survival rate of FYI, and hence a generally reduced replenishment rate (at least in the TP Drift)? Can you confirm this?**

Yes, the replenishment rate is indeed directly related to the FYI area. We have added the inference about FYI melt, and also a sentence referencing the suggested paper at the end of the paragraph. As our paper focuses on the years of 2007 and 2012, we do not want to add details of long-term trends in each region (which each experience significant variability and it would be hard to pick out one thing to zoom in on), but the new figures in the appendix show that the MYI area and volume are greatly reduced over time (including being very low in summers after 2007), and the replenishment is very close to zero in 5 of the years from 2007 onwards.

"...due to higher than average FYI melt." **(Lines 376-377)**

"This reduction in FYI available for replenishing is consistent with the reduced survival rates of FYI in marginal seas feeding the Transpolar Drift in recent years found by Krumpen et al. 2019." **(Lines 381-383)**

**Line 354-356: I guess this sentence is not needed since it is well described in 4.2.1?**

We have removed this sentence as we agree with the referee that it was superfluous here.

**Line 378: I think the Smedsrud study is not solely based on observations, but observations were used to establish a relationship between pressure gradients across Fram Strait and export?**

Yes, this is correct - the Smedsrud study uses sea level pressure gradients to derive the geostrophic winds across Fram Strait and uses these to extend the timeseries of export back in time. However, from 2004 onwards, they use observations of ice drift, so describing this study as observations when investigating the years 2011-2012 should still be relevant. We have rephrased slightly to take the emphasis off of direct observations and hope this is more suitable now:

"...when compared to the observation-based estimates of..." **(Line 425)**

**Chapter 5.2: This is all very interesting! However, I have a few questions related to the chapter**

**Line 401: Not sure if I got this correct: You mean, as the ice cover shrinks, less FYI survive the summer and hence, there is less replenishment taking place in the Central West region (and others)? Again, this would support the Krumpen story of a reduced survival rate and the timing of the drop-down in ridging and FYI survival rate (Krumpen) is about the same.**

The statement of the reviewer is right, but this is not what we meant here. With this sentence, we are not making a link between ridging and FYI survival rate, but rather the MYI area available to ridge. The FYI survival rate is discussed a bit earlier in the paper now, with reference to Krumpen et al (2019) **(Lines 381-383)**. We have now overhauled the paragraph (also in response to the comments below) and hope it is now clearer.

**And Line 400: "mostly unaffected" I would expect that ridging in the Laptev See and other Russian shelf seas went to almost zero, since MYI production zones are shifted elsewhere (north)? May be it would be a good idea to provide a Fig like Fig 6 for each section (A,B,C,D... ) in the appendix. This would also be a valuable information for other studies**

It is true that the shelf seas had less ridging after 2007, but they did not have large amounts before, and ridging was much less of a sink (as a proportion of the total losses) than the export and melt throughout the timeseries. As suggested, we have included two figures in the appendix for region-based MYI volume and area budgets, and rephrased slightly:

"In 2007, anomalously high ridging takes place first and foremost in the Central CAA and Central West regions (Figure A1f and j). After 2007, however, much less ridging takes place in the Central West region, but ridging in the other regions, where ridging contributes less to the budget in general, is mostly unaffected." **(Lines 450-452)**

**Line 401: You mean: As the MYI cover shrinks….**

We have changed this

**Line 402 – 405: Still this does not explain the stepwise decrease in ridging after 2008 (or may be I just did not get it). If this would be solely related to a shrinking MYI cover and shifting replenishment zones, it would be a gradual change, right?**

The reduction in ridging is certainly a source of interest and we agree that it requires a clearer explanation. The reasons are complex. There is a general reduction in ridging due to the reduction in MYI cover over time. But if we look at the individual regions (now Appendix A), we see that the budget of the Central West region heavily imprints on the Arctic-wide timeseries, and this is the source of the shift. Broadly, the relationship of more MYI area -> more ridging holds here, and we see an abrupt drop in the MYI area in the Central West region during 2007 and remaining low until the end of 2009. In autumn of 2009 to 2012, the MYI area in this region increases again, but ridging remains low, indicating that variability in atmospheric conditions (and therefore conditions causing ridging) also heavily affect the variability in ridging. The abrupt drop is likely amplified by the shift from MYI-dominated to mixed ice types in the region (shown in the figure below, and now also included in the manuscript), given that we assume FYI is ridged before MYI. We attempt to summarise in the text as follows:

"If we consider how ridging of MYI evolves over time in different regions, a slightly more nuanced picture appears. Ridging of MYI mainly takes place in the three central regions: Central CAA, Central Eurasian, and Central West. This is not unexpected, given the well-known main circulation patterns of Arctic sea ice (e.g. Colony and Thorndike, 1984), which generally compresses the ice against Greenland and the eastern part of the Canadian Arctic Archipelago, with a return flow through the Beaufort Gyre circulation. Regional reductions in ridging could be explained by atmospheric variability, with conditions less favourable to ridging, but also by the reduction of MYI area in regions prone to ridging and the distribution of MYI and FYI concentration within these regions. Due to our ridging assumption, the MYI fraction of the overall sea ice pack only ridges once the FYI in a given grid cell has been ridged. i.e., the potential for ridging MYI is more important if MYI is compact (with a concentration close to 100%) compared to grid cells with mixed ice types.

In 2007, anomalously high ridging takes place first and foremost in the Central CAA and Central West regions (Figure A1f and j). After 2007, however, much less ridging takes place in the Central West region, but ridging in the other regions is mostly unaffected. The reduced ridging of MYI can then be attributed to a reduction in ridging primarily in the Central West region. In this case, there is a strong reduction in both MYI area (Figure A1 i) and the average concentration of MYI after 2007 (Figure 9). This is because the band of MYI that is compacted and ridged against the Greenland and Canadian coast no longer extends far into the Central West region, meaning that there is much less MYI in that region that can be subjected to ridging. No particular change is observed for the other Central Arctic regions between the years on either side of the 2007 ridging event, either in MYI area or average concentration (Figure 9). The amount of ridging taking place in numerical models remains a challenging quantity to evaluate and an important source of uncertainty (e.g., Hunke et al., 2020). Our assumption that MYI only ridges after all FYI area has disappeared may have affected the importance of the MYI compactness in the total amount of MYI ridging, hence amplified the reduction between the difference pre- and post-2007." **(Lines 441-460, new Figure 9)**

[Figure]

**Conclusion: Line 447: "no one process stands out"… I think that the stepwise reduction of ridging somewhat stands out…**

It is true that ridging does stand out in the budget, but this cannot fully explain the decrease in MYI area (since a reduction in ridging would suggest a decreased effect on MYI, and when taking into account the overall decline in MYI available to be ridged it does not have an increased effect) and has no effect on the MYI volume. We have rephrased the sentence to make it clearer:

"…No one process by itself can explain the reduction in MYI area and volume over time". **(Line 503)**

**Line 457: "This change in behaviour related to the general reduction…". I don't think this fact has actually been explored deeply enough in the discussion to make that statement.**

We agree and have deleted the sentence.

Review 2:

**The study uses the NeXtSIM sea ice model to trace MYI area and volume and compare to satellite-based observations. MYI budgets for different regions of the Arctic are constructed and the relative contribution of source and sink terms is examined. While they find good agreement in MYI area of the model and observations w.r.t. to magnitude and trends from 2000 to 2018, the interannual variability is less well explained. They analyze 2007 and 2012 to contrast the different mechanisms.**

**I find this paper to be put together very well. The design of the study is and the key findings are presented clearly. I commend the authors for making the limitations of their results clear rather than sweeping them under the rug. I think the study makes a nice contribution towards characterizing the fate of MYI. I would have liked to see a bit more investigation on why MYI area anomalies between model and observations don't seem to match up particularly well, but maybe that's for another paper.**

We thank the reviewer for their very positive response! We agree that it is an interesting future avenue to explore the underlying reasons as to why the anomalies between the model and observations do not always match. We hope to be able to pursue further questions with this simulation in the future.

**I have some comments and suggestions for their consideration, but I think the study is a rare case of "publishable as is".**

**Well done!**

We appreciate the suggestions below and have taken them on board; please see individual comments for our responses.

**Detailed Comments:**

**Line 59: "*more comparable*"**

**comparable to what?**

By this, we meant that we use the forcing affecting the sea ice (i.e. freeze onset) as opposed to a hard-coded date to identify when MYI should be upgraded, which is therefore more in line with how passive microwave sensors differentiate between FYI and MYI than using the Arctic-wide sea ice minimum. We have tried to clarify this in the text while remaining concise:

"In our implementation we track MYI concentration, having an end-of-summer MYI source term based on the local autumn freeze onset as opposed to an arbitrary date, in a manner more directly comparable to the surface signature of MYI from satellites." **(Lines 61-63)**

**Line 103 *"do online"***

**Does this mean "in real time"?**

Yes - at model run time. We have changed "do online" to "do in real time in the model" **(Line 111)**

**Line 109:** *"average ice growth"*

**Average for a specific location or over entire domain?**

Apologies for the ambiguity here - it is the temporal average, i.e. for each day, there is a net ice growth, and this is applied to each grid cell individually. We have rephrased the text:

"This condition states that the onset of freezing following the summer melt has occurred in a grid cell element if *n* number of consecutive days of net ice growth have occurred since the summer melt began." **(Lines 116-118)**

**Line 134** *"likely an upper bound"*

**Why is this an upper bound?  If MYI is in fact still thicker than FYI  its melt rate should be higher (Bitz and Roe, 2004)? If I'm mistaken, please clarify. Maybe a brief discussion of feedbacks between thickness, ridging and melt-rates might be useful?**

We suggested this assumption is an upper bound for 3 main reasons:

- First, MYI being generally thicker than FYI, the areal melt rate of FYI should be larger (assuming this melt rate in pack ice is mostly driven by the disappearance of ice after it has thinned enough).
- The second reason concerns the volume melt rate and is closely linked to our model definition of the MYI. We have considered that ice growth due to basal freezing under MYI is not a source term of MYI, and is therefore a source of FYI. When the melt season starts, this ice at the bottom is going to be the first to melt, hence the MYI as we define it should only melt after all this ice has disappeared.
- Third, MYI generally has a thicker snow cover than FYI, which should delay the melt.

These assumptions are only valid for the difference in melt rate for a given melting season. Our understanding of Bitz and Roe (2004) is that they discuss the melt rate of MYI over longer time scales, over which the observed thinning of MYI  is faster than FYI. They call melt rate this thinning rate, and it is found to be larger for MYI as its growth is a slower process than for FYI, hence it is more difficult for MYI to readjust its thickness after important melting events.

We added these comments in the text:

"By doing so, we implicitly assume that the FYI has grown to the same thickness as the MYI by the end of the growth season and that it remains the case during the melting season. This is likely an upper bound for the melt rate of MYI, for 3 main reasons. First,  MYI being generally thicker than FYI, the areal melt rate of FYI should be larger (as neXtSIM assumes this melt rate decreases with sea ice thickness, Rampal et al., 2016). Second, MYI generally has a thicker snow cover than FYI, which should delay the melt. Third, in this study, we consider that ice growth due to basal freezing under MYI is

not a source term of MYI, and is therefore a source of FYI (see our discussion section 5.3). When the melt season starts, the MYI as we define it should only melt after all this FYI at the bottom has disappeared, which we do not factor in with our assumption on the MYI melt." (**Lines 141-150)**

**Line 165:** *grids cells with an uncertainty of < 0.02*

**I assume that's a probability? Previously probabilities were given in % (ok a bit nitpicky)**

Yes, it is a probability indeed, provided in the data. It has been changed to a % now

**Line 214.** *"the model struggles to capture the variability of the data".*

**I am glad you are pointing this out and provide the numbers! You might want to add that the two observational data sets struggle similarly to replicate each other's anomalies so that there is also some "observational" uncertainty. It seems though you have settled on OSI-SAF to be the data set of "reference" with better accuracy.**

This is true. We were hesitating about commenting on this, but have added:

"We note that the two observational datasets do not have the exact same variability, suggesting some uncertainty in the observed sub-annual behaviour." **(Lines 240-241)**

**Line 230. "*Insufficient replenishment"***

**The ERA-5 tends to be too warm in the winter, is that a possibility?**

Biases in ERA5 winter temperature may indeed limit winter ice growth, making it more likely for FYI to melt in the summer. However, given the high likelihood of compensating errors in the model (for instance the absence of an explicit melt pond scheme that may lead to underestimating the sea ice melt, or the choice of the albedo value), it is difficult to point out the responsibility of this bias for this particular case.

**Line 240:** *"strengthens our confidence that the model has a good ability"*

**So you think the error sources are therefore mostly thermodynamic? Quick scan of Boutin 2022 paper shows some numbers, is this sufficient to eliminate drift, dynamics?**

Boutin et al. show the model has good skill at capturing drift and sea ice volume transport within the Arctic Basin, better than it does at capturing the variability of winter ice growth for instance. However, it is certainly not a sufficient reason to eliminate dynamics as a source of error here, and we therefore rephrased:

"Given that the MYI can only undergo dynamical processes and melt once it has been replenished, and melt is negligible from October to April, this analysis suggests that wintertime ice transport/drift within the Arctic Basin is well-captured (which is in line with the drift and winter mass balance evaluation of the model made by Boutin et al. (2022)." **(Lines 261-263)**

**Line 249… *The model captures about 80% of the observations…***

**This is a useful analysis and the 80% number is good to know. However, a more stringent evaluation of the model skill to correctly label ice type might be relative to climatology as it isn't all that hard to correctly predict ice type for some regions (what's the accuracy of labeled anomalies in MYI area?) It is perhaps also noteworthy that the error rate doesn't seem to change as the MYI fraction decreases, which is encouraging.**

The reviewer makes a good point that some regions are more easy to predict than others due to either having very little MYI or generally having MYI cover year-round. We have included below the same analysis as in Figure 5 but for each region. It shows that some regions do better or worse than others, particularly in the years that the full Arctic comparison highlights as less good. We have added the average percentages that describe these plots in the text, in a new paragraph at the end of the evaluation section, to explain this, and have moved the description of the Arctic sub-regions into this paragraph to detail where the sub-regions come from:

"To further investigate the quality of the results, we look at the spatial agreement in individual sub-regions of the Arctic (Figure 8). We define these regions in a way similar to Boutin et al (2022) and Ricker et al (2021). We use four of the outer regions of Boutin et al (2022) and Ricker et al (2021), corresponding to the Chukchi, Beaufort, and East Siberian and Laptev seas (the latter two of which are combined as they contain very little MYI). We sub-divide the Central Arctic region of Ricker et al (2021) into three: the "Central CAA", corresponding to the portion north of the Canadian Arctic Archipelago, the "Central Eurasian", covering the Eurasian Basin portion of the Central Arctic, and the "Central West", covering the western portion. We find that regions of mainly MYI-free or full MYI cover do very well (95% ± 3% for the East Siberian and Laptev seas, 94% ± 8% for the Central Canadian Arctic Archipelago region) while the other regions with more mixed ice cover still perform reasonably (77% ± 13% for the Chukchi region, 70% ± 11% for the Beaufort region, 67% ± 9% for the Central East region, and 76% ± 17% for the Central West region). There are two periods that fall below a 70% match on the pan-Arctic scale: the winter of 2001 to spring 2003, and winter of 2016 to autumn of 2018, which correspond to periods of large uncertainties in the observations (autumn 2001) or too much ice loss in the summer and therefore too little going into the autumn. The performance of the Central East and Central West sub-regions are most affected by these years." **(Lines 276-288)**

[Figure]

**lIne 276:** *remains large*

We have deleted "still"

**See also Moore et al. 2022**

We have added a reference to the Moore et al (2022) paper, but at the end of the paragraph about exports instead as it discusses redistribution of ice leading to melt:

"Within the Arctic domain, some regions experience notable net loss of MYI due to export (such as the Siberian and Laptev seas and the Central West region), while the Beaufort region experiences a large net gain (Figures 8, A1, and A2), the latter providing the source to the large melt that occurs there \citep{Moore2022}." **(Lines 318-320)**

**Line 285** *" relative MYI area loss"*

**Maybe better to say "increased  contribution to the total MYI loss". I was scratching my head a bit what "relateive MYI loss meant". The sentence could use some rephrasing.**

Apologies for the unclear phrasing. We agree and have rephrased as suggested  **(Line 317)**

**Line 265…** *which represents over one third…*

**Those are good numbers for perspective. Are those derived from the budget "nets" (sum of nets) or a trend like in figure 3b?**

They are computed from differencing the first area (/volume) and the last area (/volume) of the timeseries, so essentially the sum of the nets. We have rearranged the sentence to emphasise the "total" part:

"MYI area declines over 2000--2018, with a total net loss of ~1000 x $10^3$ km$^2$ , which represents over one third of the total Arctic MYI area in 2000." **(Lines 295-297)**

**Line 287 … *"there is no trend in any of the source and sink terms"***

**As well as their sum ('net')?**

We had a similar comment from the other reviewer, and would like to apologise for how this is phrased. We did various tests of trends and significance and unfortunately did not summarise it well. If we sum the contribution from each process, this net gain/loss does not have a trend. But on an Arctic-wide scale, if we look at each process separately, there is a significant trend in some of the contributors to the budget - namely ridging for area, and melt, replenishment and the sum of the sink terms for volume. These significant trends would initially suggest that the contributions from those terms are reducing over time.  In the case of the replenishment, a reduction in replenishment could lead to a reduction in MYI, but the trend in volume replenishment is not large enough to explain the MYI losses. In the case of the sink terms, this does not explain the reduction in MYI since it would suggest less contribution from sink terms than before, which is counter-intuitive as MYI area is reducing over time. This is misleading because this computation does not consider the fact that there is less MYI to melt over time. Therefore the trends on the raw values do not give us much information on how the processes are changing. We concluded it was more relevant to analyse trends of the processes divided by the amount of MYI available to be depleted, and therefore recomputed the trends on the processes divided by the amount of MYI area or volume on January the 1st (as we also do for the anomalous years in the next paragraph), and this time there are no significant trends in any process. We have added a few sentences to attempt to explain this thought process in a concise way, and hope that it is now clearer.

"However, there is no trend in any of the source or sink terms that can explain the losses of MYI area and volume; indeed, apart from a reduction in volume replenishment, the other significant trends (ridging and volume melt) show a decrease in contribution over time rather than increase. The volume replenishment reduction trend is not large enough to account for the loss of MYI volume. The reduction in the sink terms is misleading, it is driven by the fact that MYI area is reducing over time and not due to a change in the processes themselves. We conclude we need to account for MYI area and volume decline in our computation, which we do by recomputing trends for the loss/gains as a proportion of the respective MYI area or volume on the 1$^{st}$ January each year. We then find there is no trend in any process." **(Lines 330-336)**

**I'm also not quite understanding the argument about the "episodic imbalance".  If the net is negative, even without a temporal trend, that will yield declining MYI area and volume. Is the idea that the "mean net negative" over the period is the result of a few years? It might be useful to try**

**to quantify this X % of "imbalance" over the 18 year period arises from Y years? Or with without years X,Y,Z, the net would be balanced?**

We are sorry for the ambiguity here. Yes, it is a result of a few years. What we mean by an episodic imbalance is that the net switches between positive and negative, and if 2007 and 2012 are removed from the timeseries then the MYI area contributions area actually positive, and the MYI volume losses are greatly reduced by removing 2012 (note that 2007 results in an increase in MYI volume). We have explained this in the text, and additionally moved the statement about episodic imbalance to follow this explanation, and hope it is now clearer:

"To put these two years in the context of the overall net losses over the time period, we find that if we remove the total contributions of these years to the budget, the remaining years actually result in a net gain of MYI area. It is notable that these two modelled extreme years coincide with years that set new record low total Arctic sea ice extents (e.g. Parkinson and Comiso, 2013). For MYI volume, two years experience extreme loss, namely 2012 and 2016. Removing these years results in a net gain of MYI volume. Therefore, the negative MYI area and volume trends are associated with an episodic imbalance between the different loss terms and replenishment rather than a constant net loss. To better understand the drivers of MYI loss, we therefore focus on 2007 and 2012, excluding 2016 since the agreement of MYI extent between model and observations is less good in that year." **(Lines 345-350)**

**Line 307 "** *two anomalous years 2007 and 2012***"**

**I don't really see this in Figure 6a? Maybe 6b/d but also doesn't exactly jump out ( I suspect within 1 sigma?). 2007 and 2012 are of course notable minima in September ice cover, maybe that's the better motivation. I do like the "case study" analysis for those two years and haven't seen the 2007 case nicely documented as here.**

It is perhaps a little difficult to see this in the timeseries because the seasonal cycle is clouding the signature. We have pointed out in the text now that the red line on Figures 6b and 6d provide an indication of these years, and added a reference to those being record sea ice minima to help motivation.

"It is notable that these two modelled extreme years coincide with years that set new record low total Arctic sea ice extents (e.g. Parkinson and Comiso, 2013)." **(Lines 347-348)**

We have also rearranged the first sentence of the discussion section to allow for the fact that we now introduce the sea ice minima into the paper earlier:

"Given that the extreme years of 2007 and 2012 are not only years of extremely low MYI extent, but also years of extreme September sea-ice extent minima (Stroeve et al., 2008; Comiso, 2008; Parkinson and Comiso, 2013), they are of interest for both the evolution of MYI and the evolution of Arctic sea ice in general." **(Lines 386-388)**

**I also wonder a bit how the analysis would look like if you defined years not from January 1 but as "ice years" from October 1 through September 30th . I wonder if that would not show the years**

**a bit more "anomalous" as processes prior and subsequent to annual minima might be better separated (haven't thought this through, just wondering)**

We experimented with different start and end dates for a "year" for the budget, noting that wherever the cut-off is put, a process will be interrupted. A test of October-September suggests that (perhaps unsurprisingly) the only terms that are slightly different are export and ridging. These have a good agreement between the January-December and October-September budgets (correlation > 0.8) with only a few years slightly offset in the timeseries. The most notable is ridging in 2008, which in the January-January timeseries reduces significantly compared to October 2007-September 2008. This is because some of the extreme ridging that takes place in 2007 occurs in the autumn, but we note that apart from this event, it seems that ridging and export from October to December do not vary much year to year. Therefore, it does not significantly affect the analysis to transfer the autumn part of the budget from one calendar year to the next. As a result, we prefer to keep it as January to January as it is much easier to describe a given year in the text rather than spanning two years, and it also gives us the use of the whole of 2000 and 2018 in the analysis.

**Line 363 *so the 212 August Storm***

**I don' t think the storm was mentioned before? It would probably be helpful to add the information that this storm occurred and provide a reference (I see Parkison and Comiso 2013 already cited could cited again here, also, Simmonds and Rudeva, 2012, Zhang et al. 2013, Stern et al. 2020)**

Yes, you are right, we apologise for this! We have introduced the storm now at the beginning of the paragraph.

"The enhanced melt in 2012 has been documented as a big driver of the total sea ice minima in that year, due to an August cyclone occurring on the backdrop of a thinning ice pack (Parkinson and Comiso, 2013; Lukovich et al., 2021) and near-surface ocean heat (Zhang et al., 2013)." **(Lines 401-403)**

And also referenced the storm **(Lines 409-410)**.

**Line 367 *well with observations, then there is***

**I think "then" is a typo and can be deleted**

Yes - we have deleted this

**Line 409*: surface signature in MYI***

**Does this refer to the microwave signature? Sticklers might argue that this isn't strictly related to surface alone. Maybe "remote sensing" signature?**

We have taken the suggestion **(Line 465)**

**Line 434 the modeled ice drift is very good**

**This isn't really shown in the paper, is it? There is a reference to Boutin 2022 which I only scanned and which has some numbers. It should be made clearer what is found in the present paper that may draw on conclusions from other papers. I guess the message is that ice-drift was found to be accurate by Boutin et al 2022 and isn't the source of the model error. The finding in the present paper is that the model MYI extent mismatch in the fall is the primary source of error? As I said above, that could probably be worked out more clearly and the specific open questions regarding the source of the error could be more definitive (thermodynamic physics, forcings?). From my quick scan of Boutin et al. 2022, it appears that while drift stats look good, I wonder how well this translates to eliminating dynamics as a significant error source?**

It is true that this is not directly shown in the paper. With this statement, we were more meaning that if the autumn replenishment is reasonable, then the evolution during the winter is good, which can only be due to good dynamics (or strong melt, which is not a sink term in winter). But this meaning is perhaps clouded by discussing the model drift, so we have removed it.

"The model generally agrees well with multiple datasets, both in integrated amount of MYI extent and its spatial distribution, and the spatial distribution of MYI is good, as long as the minimum ice extent in the autumn is reasonably well captured." **(Lines 489-491)**

To help the reader, we have also introduced better what is discussed in Boutin et al., 2023 in the model setup **(Lines 89-95)** and what are the main conclusions of their study that are relevant to ours.

**The NeXtSIM model seems to be particularly well suited for the assimilation of sea ice motion, isn't it? Using an equivalent analysis of MYI budgets with assimilation could isolate the effect of inadequately simulated sea ice motion (winds/drag) on the results (future work!)**

We had not thought about that, that is a good idea!

**Suggested References:**

Moore, G. W. K., Steele, M., Schweiger, A. J., Zhang, J., & Laidre, K. L. (2022). Thick and old sea ice in the Beaufort Sea during summer 2020/21 was associated with enhanced transport. Communications Earth & Environment, 3(1), 198. doi:10.1038/s43247-022-00530-6

Simmonds, I., & Rudeva, I. (2014). A comparison of tracking methods for extreme cyclones in the Arctic basin. Tellus Series a-Dynamic Meteorology and Oceanography, 66. doi:10.3402/tellusa.v66.25252

Stern, D. P., Doyle, J. D., Barton, N. P., Finocchio, P. M., Komaromi, W. A., & Metzger, E. J. (2020). The Impact of an Intense Cyclone on Short-Term Sea Ice Loss in a Fully Coupled Atmosphere-Ocean-Ice Model. Geophysical Research Letters, 47(4), e2019GL085580. doi:https://doi.org/10.1029/2019GL085580

[revised manuscript text omitted]

---

## Author Response (AR2)

Dear Editor,

This note is to explain a few minor changes to the manuscript that have been carried out after the response to reviewers process was complete.

In the round of revisions, reviewer 1 had the following comment:

**Line 191: May be you can get in touch with the producer and ask what the reason may be?**

This was in reference to the following sentence in the manuscript:

**"While such an increase in MYI extent can occur mid-winter due to diverging ice drift, neither the NSIDC_age data nor the model show such an extensive increase, potentially indicating random errors in the CDR dataset."**

We followed the advice of the reviewer and contacted the producer of the CDR dataset about why "such an extensive increase" was appearing in their data. At the time of completing our reviews, we had not had a response, so could not add details in the manuscript to inform readers as to why there was a discrepancy. However, after the review process was complete, we received an answer to our query. In order to be able to answer the reviewer's original comment, and to ensure the manuscript was as informative and accurate as possible, we requested to be allowed to incorporate the response to our query into the manuscript. This involved two very minor changes to the manuscript (along with one in the data availability section and one in the references), which do not in any way change the message in the text but rather serve to clarify the wording and provide some reason as to the difference we see.

Change 1 (Lines 167-168)

"We use the Ocean and Sea Ice Satellite Application Facilities (OSI-SAF)  Climate Data Record ice type product (Aaboe et al., 2019)"

has been changed to

"We use version 1 of the sea ice type Climate Data Record from the Copernicus Climate Change Service (C3S) Climate Data Store (CDS) (2020), hereafter CDR"

*The dataset is the same, we have just updated the reference and made the wording more specific.*

Change 2 (Lines 212-216)

"potentially indicating random errors in the CDR dataset"

has been changed to

"Note that in this study we use version 1 of the dataset, which is now deprecated and fully replaced by an upgraded version. The recently released version 3 of the dataset has major improvements relative to version 1, including both gap filling (bringing it to a level 4 product)

and correction schemes based on temperature and ice drift information to correct for mis-classifications and to improve the early autumn classification (S. Aaboe, personal comms)."

Change 3 (Data availability, lines 466-467)

"The ice type product from the OSI-SAF Climate Data Record is available at https://thredds.met.no/thredds/c3s/c3s.html (last visited June 2020)."

has been changed to

"The ice type product Climate Data Record (version 1) from the Copernicus Climate Change Service (C3S) Climate Data Store (CDS) (2020) is available at https://thredds.met.no/thredds/c3s/c3s.html (last visited June 2020)."

Change 4 (reference, lines 560-561)

We added a reference to the ice type dataset, in addition to the original reference to the documentation:

"Copernicus Climate Change Service (C3S) Climate Data Store (CDS): Sea ice edge and type daily gridded data from 1978 to present derived from satellite observations, version 1.0, https://doi.org/10.24381/cds.29c46d83, [Accessed on 30-06-2020], 2020."

We are grateful for having the opportunity to make these small but important changes to the text.

Kind regards,
Heather Regan